# Mutant p53 gains oncogenic functions through a chromosomal instability-induced cytosolic DNA response

Mei Zhao [1,12], Tianxiao Wang[1,2,12], Frederico O. Gleber-Netto [1], Zhen Chen[3], Daniel J. McGrail [4,9,10], Javier A. Gomez [5], Wutong Ju [1], Mayur A. Gadhikar[1], Wencai Ma[6], Li Shen[6], Qi Wang [6], Ximing Tang[7], Sen Pathak[8], Maria Gabriela Raso [7], Jared K. Burks [5], Shiaw-Yih Lin [4], Jing Wang [6], Asha S. Multani[8], Curtis R. Pickering[1,11], Junjie Chen [3], Jeffrey N. Myers [1] ✉ & Ge Zhou [1] ✉

Inactivating *TP53* mutations leads to a loss of function of p53, but can also often result in oncogenic gain-of-function (GOF) of mutant p53 (mutp53) proteins which promotes tumor development and progression. The GOF activities of *TP53* mutations are well documented, but the mechanisms involved remain poorly understood. Here, we study the mutp53 interactome and find that by targeting minichromosome maintenance complex components (MCMs), GOF mutp53 predisposes cells to replication stress and chromosomal instability (CIN), leading to a tumor cell-autonomous and cyclic GMP–AMP synthase (cGAS)-stimulator of interferon genes (STING)-dependent cytosolic DNA response that activates downstream non-canonical nuclear factor kappa light chain enhancer of activated B cell (NC-NF-κB) signaling. Consequently, GOF mutp53-MCMs-CIN-cytosolic DNA-cGAS-STING-NC-NF-κB signaling promotes tumor cell metastasis and an immunosuppressive tumor microenvironment through antagonizing interferon signaling and regulating genes associated with pro-tumorigenic inflammation. Our findings have important implications for understanding not only the GOF activities of *TP53* mutations but also the genome-guardian role of p53 and its inactivation during tumor development and progression.

The tumor suppressor p53 is among the most important factors that preserve genomic stability and integrity[1]. Early studies of p53 function and the consequences of *TP53* mutation demonstrated that p53 regulates DNA-damage-induced cell cycle checkpoints and inactivation of which enables genomic instability[2–4], indicating that p53 acts as the "guardian of the genome" which prevents cells from developing potentially tumor-promoting mutations and abnormal genomes[5]. A recent study in a mouse model of pancreatic ductal adenocarcinoma further showed that the inactivation of p53 does not merely open a gateway to genetic chaos but, rather, can enable a predictable pattern of cancer genomic development (i.e., accumulation of deletions, genome doubling, and the emergence of gains and amplifications), suggesting that p53 and its inactivation play a deterministic role in the ordered development of CIN in cancer[6]. Although in vivo evidence has indicated that p53 suppresses tumorigenesis by inducing a set of transcriptional programs that prevent the proliferation of cells and that ongoing p53 inactivation is needed to sustain malignant disease[4,7–11], this latest study suggested that p53 inactivation is not sufficient in and of itself for malignant transformation; the acquisition of recurrent chromosomal copy number alterations (CNAs) is also

required[6]. Because CNAs/aneuploidy achieved by CIN cause a wide range of tumorigenic consequences[12], it is important to further understand whether the oncogenic phenotypes associated with CIN— altered proliferation[13], cell survival[14], drug resistance[15], metastasis[16], immune evasion and resistance to immunotherapy[17], and shorter patient survival in many cancers[18,19] are attributable to altered gene expression programs due to the gain and/or loss of genes, or whether CIN itself has also a direct impact on the development of these onco- genic phenotypes. Similarly, a better understanding of how p53 inac- tivation enables the development of CIN is also critical. That is, it remains unclear whether p53 inactivation-induced CIN is due only to the loss of the transcription programs of wild-type p53 (wtp53) or whether other mechanisms are also involved.

Inactivating mutations in *TP53* are the most common cancer dri- ver mutations and present a route to malignant transformation in more than half of all human cancers[20–22]. Although inactivating muta- tions of *TP53* lead to loss of function (LOF) of wtp53 and/or dominant- negative mutp53s, studies using mouse models clearly indicate that some mutp53s can confer oncogenic GOF activities to promote tumorigenesis and metastasis[23,24]. So far, mutp53 GOF activities, such as the promotion of cell proliferation, survival, invasion, cancer- promoting metabolism, drug resistance, and pro-tumorigenic inflam- mation, have been observed in in vitro and in vivo experimental models, and many GOF mechanisms, such as mutp53 interaction with transcription factors and/or cellular proteins that impact transcription programs and cellular functions, have been proposed[25–33]. However, these proposed mechanisms are often context-dependent, and many mutp53-mediated GOF phenotypes vary across experimental settings (e.g., in different cells or clones). Interestingly, in striking contrast to all of the previously proposed mechanisms, a recent study demonstrated that expression of GOF mutp53 has no close causal relationship with GOF phenotypes including gene expression changes, enhanced pro- liferation, tumorigenicity, tumor growth, metabolic alterations, and drug resistance. Moreover, that study demonstrated that GOF phe- notypes are only associated with aneuploidy in cells with mutp53[34]. This finding strongly suggests that the consequences of CIN in mutp53 cells may contribute to many previously reported mutp53 GOF phe- notypes. Given the strong propensity for cells with mutp53 to become aneuploid and the overlap of oncogenic phenotypes related to mutp53 GOF and aneuploidy, it has been suggested that GOF phenotypes identified in mutp53 models must be carefully validated relative to corresponding chromosomal changes[34]. Therefore, while this study highlighted the importance of CIN in mutp53 GOF-associated pheno- types and suggested that acquisition of aneuploidy may be a unifying mechanism that accounts for many context-specific GOF phenotypes previously attributed to mutp53 proteins, the underlying mechanisms that link GOF mutp53 and CIN still remain unclear.

Although LOF mutations of p53 impair cell cycle checkpoints and DNA repair, which can lead to the development of CIN[2–4], accumulating evidence has also shown that many mutp53s not only lose the genome- guardian role of wtp53 but also gain oncogenic functions to promote CIN through targeting various proteins involved in genome stabilization[26]. Six MCMs form a MCM2-7 hexameric complex. As the key DNA replication regulator, the MCM2-7 complex forms the DNA replication licensing system, which assembles the replication machinery at replication origins with the origin recognition complex during replication initiation. This complex forms the catalytic core of the Cdc45/Mcm2-7/GINS helicase that unwinds DNA during elongation[35]. Therefore, disruption and/or dysregulation of the MCM2- 7 complex predisposes cells to replication stress, CIN, and the devel- opment of cancers[36–39]. Here, we examined the mutp53 interactome and identified MCMs as the protein targets through which mutp53 exerts its GOF activities to predispose cancer cells with mutp53 to replication stress and CIN. In turn, these events lead to a tumor cell- intrinsic cytosolic DNA response involving cGAS-STING-dependent activation of downstream NC-NF-κB signaling, which promotes tumor metastasis and tumor immunosuppression.

## Results

### GOF mutp53 interacts with MCMs

To identify the mutp53 interactome, we expressed G245D mutp53 in p53-deficient head and neck squamous cell carcinoma (HNSCC) UM- SCC-1 cells and used them for quantitative proteomic analyses using both stable isotope labeling with amino acids in cell culture (SILAC) and immunoprecipitation (IP) with 2 different p53 antibodies (DO-1 and Pab240) (Supplementary Fig. 1a). After purification, the protein complexes were quantitatively analyzed by liquid chromatography coupled with tandem mass spectrometry. In 8 independent experi- ments, we identified 33 proteins that interacted with G245D mutp53 in UM-SCC-1 cells (Supplementary Data 1). Many of them, such as the p63[40], FAM83A[41], FUBP1[42], TGM2[43], SNRPN[44], MCM5[45], and MCM7[46], were previously shown to interact with wtp53 and/or mutp53s, which validated the reliability of our approach. Metascape enrichment analysis[47] of purified G245D mutp53-interacting proteins showed that the gene ontology (GO) term "regulation of DNA replication initiation" involving MCM5, MCM7, and WRNIP1 was significantly enriched (Fig. 1a and Source Data to Fig. 1a), which strongly suggests that mutp53s play an important role in regulation of DNA replication initiation.

To investigate the mechanisms involved, the specific interaction of MCM5 and MCM7 with mutp53s was further validated by reciprocal IP assays using cell lines with various endogenous p53 mutations in the presence or absence of interferon γ (IFN-γ), which was previously shown to promote the interaction of MCM5 and Stat1[48] (Fig. 1b, c and Supplementary Fig. 1b–e). Furthermore, our IP assay of cells co- transfected with exogenous wtp53 or mutp53s and MCM5 demon- strated that (1) mutp53s (i.e., R273H, G245D, R175H) bound to MCM5 (Fig. 1d, lanes 4–6 and Fig. 1e); (2) wtp53 and mutp53 R248Q did not bind (or weakly bound) to MCM5 (Fig. 1d, lanes 3, 7 and Fig. 1e), sup- porting the idea that wtp53 and these various mutp53 proteins have different properties[28]; (3) both N-terminal transactivation domain- and DNA-binding domain-deleted R273H mutp53s (i.e., R273H-ΔTA and R273H-ΔDBD) bound to MCM5 (Fig. 1d, lanes 8 and 11), whereas C-terminal domain deletion of R273H mutp53 resulted in loss of MCM5-binding activity (Fig. 1d, lane 10), suggesting that R273H mutp53 binds to MCM5 through its C-terminal domain; (4) although R273H/L and several other mutp53s also interacted with other com- ponents of the MCM2-7 complex (e.g., MCM2, MCM3, and MCM7), they bound most strongly to MCM5 (Fig. 1f, g and Supplementary Fig. 1c–e). Finally, although R248Q mutp53 did not bind to MCM5 (Fig. 1d, lane 7 and Supplementary Fig. 1f, lane 6), our IP results using UM-SCC-1 cell line expressing R248Q mutp53 showed that it strongly bound to MCM7 (Supplementary Fig. 1g, lane 4), suggesting that dif- ferent mutp53s interact with the components of the MCM2-7 complex differently. Collectively, our results demonstrate that mutp53s interact with MCMs.

### GOF mutp53 predisposes cells to replication stress and CIN through MCM5

MCM5 and MCM7 are two of the 6 components of the MCM2-7 het- erohexameric complex. Given that MCM2-7 forms the DNA replication helicase system that is important for high-fidelity DNA replication and genomic stability[35–39,49–51], we examined the functional role of the mutp53-MCM5 interaction in the development of replication stress and CIN. To this end, we introduced the expression of exogenous wtp53 or GOF mutp53s in *TP53*-knockout, immortalized, non- transformed human normal epithelial hTERT HAK cl41 cells and p53- deficient UM-SCC-1 cancer cells. In response to a low concentration (100 μM) of hydroxyurea (an inhibitor of the enzyme ribonucleotide reductase that depletes the pool of available deoxyribonucleotides and induces replication stress), cells expressing G245D, R273H, or

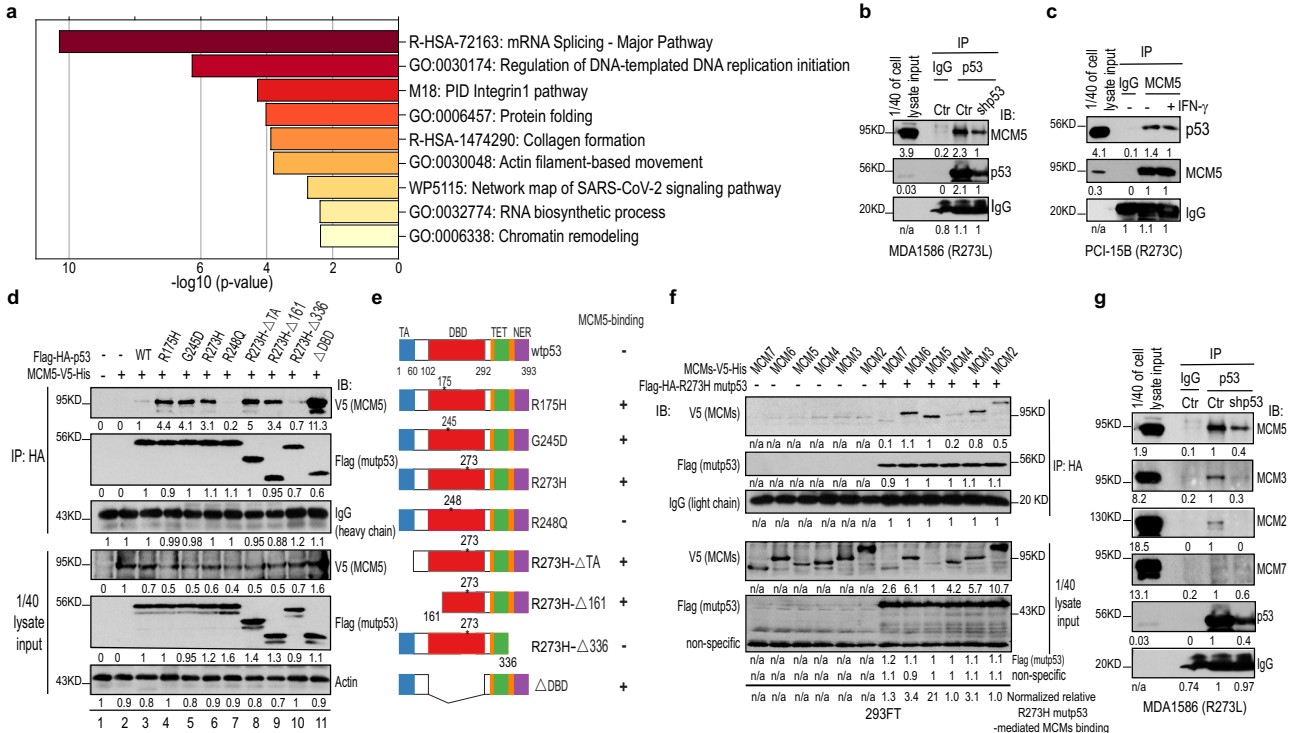

**Fig. 1 | mutp53 interacts with MCMs. a** Bar graph of enriched terms across input gene lists (colored by *P*-values) from SILAC/immunoprecipitation (IP) purification and mass spectrometry of G245D mutp53 interactome in UM-SCC-1 stable cells from Metascape analysis. Pairwise similarities between any two enriched terms were computed based on a Kappa-test score. The similarity matrix was then hierarchically clustered and a 0.3 similarity threshold was applied to trim the resultant tree into separate clusters. The most significant (lowest *P*-value) term within each cluster was chosen to represent the cluster in the bar graph. **b** p53 antibody (DO-1) IP/Western blot analysis using MDA1586 cell lysates and MCM5 antibody. Cells with mutp53 knockdown (shp53) were used as the control to evaluate the specificity of the IP. Normalized quantitative results (densitometry) calculated using NIH Image J software are shown under each blot. IB, immunoblotting. **c** MCM5 antibody IP/ Western blot analysis using PCI-15B cell lysates and p53 antibody. IFN-γ, 200 IU, 30 min. **d** HA antibody IP/Western blot analysis of HEK293-FT cells co-transfected with V5-tagged MCM5 and Flag-HA-tagged wtp53 or various mutp53s. **e** Summary of the results from **d**. **f** HA antibody IP/Western blot analysis of HEK293-FT cells co-transfected with Flag-HA-tagged R273H mutp53 and various V5-tagged MCM components. **g** p53 antibody IP/Western blot analysis using MDA1586 cells and various MCM antibodies. Shp53, cells with mutp53 knockdown. Source data are provided as a Source Data file.

R248Q mutp53, but not those expressing wtp53 or the C-terminus truncated Δ336 mutp53 that does not bind to MCM5 (Fig. 1d, lanes 3 & 10), exhibited greater chromatin accumulation of both mutp53s and phosphorylated and/or total replication protein A 32 kd subunit (RPA32), a marker of replication stress, than did the control cells (Fig. 2a, lanes 12 vs. 8 and 10; Fig. 2b, lanes 18 vs. 14, 22 vs. 14; Supplementary Fig. 1h, lanes 4 and 6 vs. 2 and Supplementary Fig. 1i, lanes 6 vs. 4). Consistent with this, more nuclear RPA32 foci were seen in mutp53-expressing hTERT HAK cl41-p53KO-c1 cells treated with hydroxyurea than in control cells without mutp53 expression (Supplementary Fig. 1j). Given the strong interaction between MCM5 and G245D or R273H/L mutp53 (Fig. 1f, g and Supplementary Fig. 1f), we focused the following studies on investigation of the functional role of the interaction between MCM5 and G245D or R273H/L mutp53 in regulation of DNA replication, and then we further explored its consequential biological implications.

In line with the strong interaction of MCM5 and R273H mutp53, R273H mutp53-induced chromatin accumulation of RPA32 in response to hydroxyurea treatment was efficiently suppressed by over-expression of MCM5 (Fig. 2b, lanes 20 vs. 18, 24 vs. 22; Supplementary Fig. 1k, lanes 16 vs. 14 and Supplementary Fig. 1l, lanes 16 vs. 14) but not by expression of MCM2 (Supplementary Fig. 1l, compare lanes 16 vs. 14 to 8 vs. 6) that has weak R273H/L mutp53 binding (Fig. 1f, g). To further test the specificity of MCM5's impact, we generated an inactive MCM5 mutant (R732A & K734A) that was unable to interact with other components of the MCM2-7 complex[48], such as MCM3 (Supplementary Fig. 1m, lanes 5 vs. 6) but still maintained R273H mutp53-binding

activity (Supplementary Fig. 1n, lanes 8 vs. 7). This mutant MCM5 exhibited greater inhibition of R273H mutp53-induced chromatin accumulation of RPA32 than did the wild-type MCM5 in the presence of hydroxyurea (Supplementary Fig. 1o, compare lanes 10 vs. 6 to 8 vs. 6). This was because, in addition to inhibition of mutp53, over-expression of the wild-type MCM5 subunit alone likely disrupted the stoichiometry of the MCM2-7 complex (i.e., dominant-negative effect), thereby impairing MCM2-7 complex function and predisposing cells to replication stress (Supplementary Fig. 1k, l, lanes 11 and 12 vs. 9 and 10 and Supplementary Fig. 1o, lanes 4 vs. 2), whereas expression of R732A and K734A mutant MCM5 did not impact the MCM2-7 complex, but only bound to and inhibited mutp53.

In further support of the role of mutp53-MCM5 signaling in pre-disposing cells to replication stress, the knockdown of endogenous mutp53 in MDA1586 (R273L mutp53) and PCI-15B (R273C mutp53) cells led to much less chromatin accumulation (Fig. 2c, lanes 4 vs. 2 and Supplementary Fig. 2a, lanes 4 vs. 2), fewer nuclear foci of RPA32 (Supplementary Fig. 2b), and fewer stalled and more newly initiated DNA forks in the DNA fiber assay (Supplementary Fig. 2c–e) than were seen in parental control cells treated with hydroxyurea. However, further knockdown of MCM5 (Supplementary Fig. 2f, g) rescued RPA32 chromatin accumulation (Fig. 2c, lanes 6 vs. 4, 8 vs. 4 and Supplementary Fig. 2h, lanes 6 vs. 4, 8 vs. 4), formation of RPA32 nuclear foci (Supplementary Fig. 2i), and stalled DNA forks (Supplementary Fig. 2c–e) under the same condition.

In line with these results, more chromosomal abnormalities were seen in immortalized, non-transformed human normal epithelial hTERT

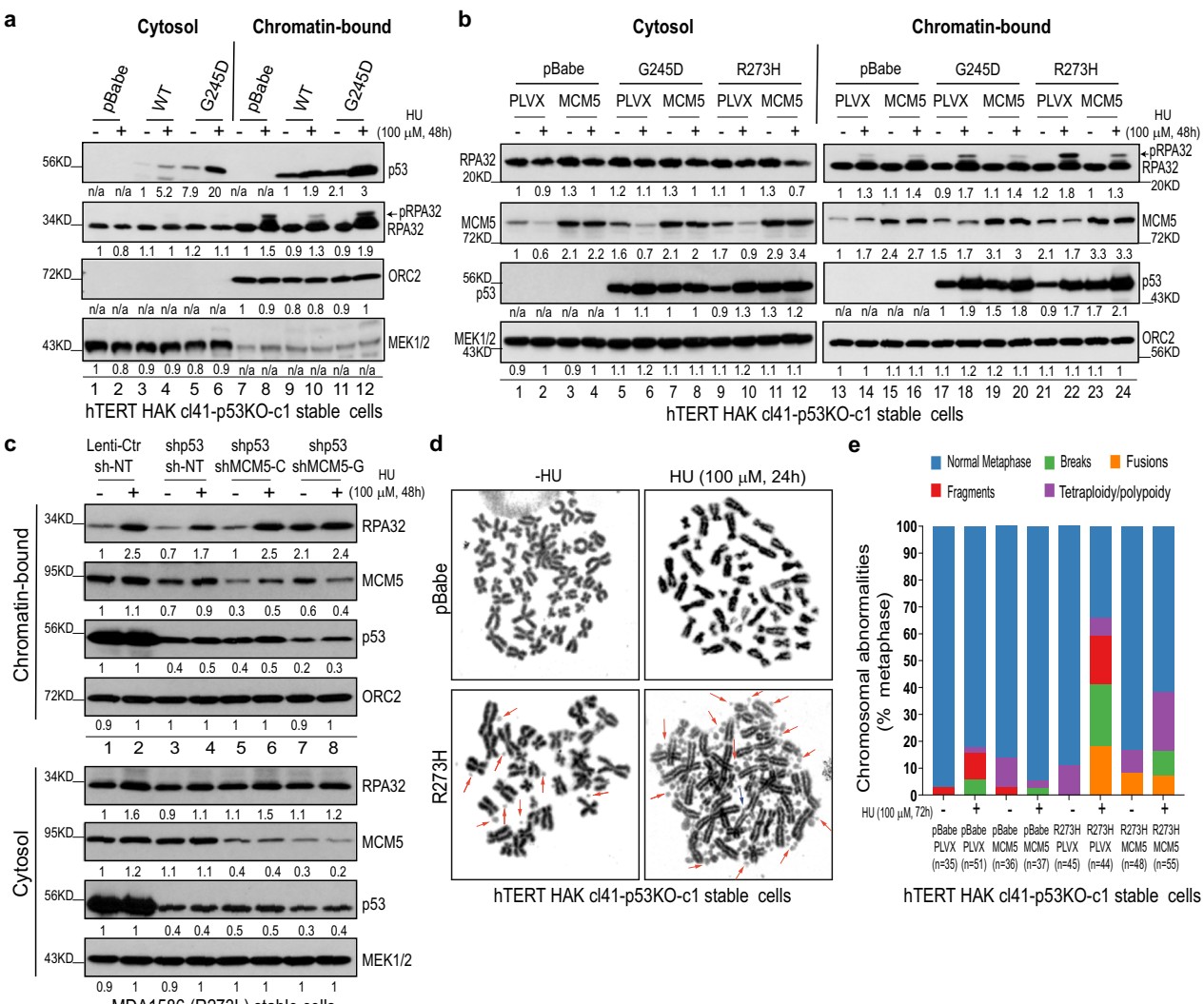

**Fig. 2 | GOF mutp53 predisposes cells to replication stress and CIN through MCM5. a–c** Western blot analyses using the cytosol and chromatin-bound fractions from the indicated stable cell lines. Overexpression in stable cell lines was established by sequential retroviral (pBabe-wt/mutp53s) and lentiviral (pLVX-MCM5) infections (**a, b**). Gene downregulation in cell lines was established by sequential lenti-shp53 and two independent doxycycline (DOX)-inducible-shMCM5 lentiviral infections (shMCM5-C or -G) (**c**). In **c**, cells were cultured in a medium with DOX (200 ng/mL) for 48 to 96 h before further hydroxyurea (HU) treatment. NT, non-target. pBabe and pLVX were the control empty vectors. pRPA32, phospho-

RPA32. MEK1/2, mitogen-activated protein kinase 1/2. ORC2, origin recognition complex 2. **d** Examples of the metaphase spreads of hTERT HAK cl41-p53KO-c1 stable cells in the absence or presence of HU treatment. Representative chromosomal abnormalities are marked by arrows (red, fragments/double minutes; light blue, fusions). **e** Summary of chromosomal abnormalities (breaks, fragments, fusions, and tetraploidy/polyploidy) in the metaphase spreads of hTERT HAK cl41-p53KO-c1 stable cells in the absence or presence of HU treatment. Source data are provided as a Source Data file.

HAK cl41-p53KO-c1 cells expressing R273H or G245D mutp53 than in control cells, especially in the presence of hydroxyurea (Fig. 2d, e and Supplementary Fig. 2j, k). In contrast, the downregulation of endogenous R273L mutp53 in MDA1586 cells reduced the chromosomal abnormalities in response to hydroxyurea treatment (Supplementary Fig. 2l, m). Most importantly, these mutp53-induced chromosomal abnormalities were further inhibited by further MCM5 overexpression (Fig. 2e). Taken together, our results not only show that GOF mutp53 predisposes cells to replication stress and CIN but also strongly suggest that this predisposition is due to mutp53 and MCM5 interaction that leads to functional disruption of MCM2-7 complex.

### GOF mutp53-MCM5-mediated replication stress and CIN stimulate cytosolic DNA and 2'3'-cyclic GMP–AMP (cGAMP) accumulation

Replication stress and CIN induced by chromosomal missegregation can activate innate immune signaling through the

introduction of genomic DNA into the cytosol and activation of the cGAS-STING pathway[16,52]. In agreement with this, we found that overexpression of G245D or R273H mutp53 in p53-null UM-SCC-1 cells (Supplementary Fig. 3a) resulted in more accumulation of both cytoplasmic double-stranded DNA (dsDNA) and single-stranded DNA (ssDNA) than in control cells in either the absence or presence of hydroxyurea (Fig. 3a, b and Supplementary Fig. 3b–h). Consistent with its "dominant-negative effect" on the MCM2-7 complex, MCM5 overexpression alone in the absence of mutp53 increased the accumulation of cytosolic dsDNA (Fig. 3a, b) in response to hydroxyurea treatment in UM-SCC-1 cells, but overexpression of MCM5 in the presence of mutp53 inhibited mutp53-induced cytosolic dsDNA and ssDNA accumulation under the same conditions (Fig. 3a, b and Supplementary Fig. 3c–e, g, h). Furthermore, knocking down mutp53 in MDA1586 or PCI-15B cells significantly decreased both cytosolic dsDNA and ssDNA accumulation, but this decreased DNA accumulation was rescued by

further knocking down MCM5 expression in these cells (Fig. 3c, d and Supplementary Fig. 3i–n). Consistently, overexpression of R273H mutp53 in p53-null UM-SCC-1 cells resulted in more intracellular cGAMP accumulation than in the control cells in either the absence or presence of hydroxyurea (Supplementary Fig. 3o). Moreover, MCM5 overexpression alone in the absence of mutp53 increased the accumulation of cGAMP in response to hydroxyurea

treatment, but overexpression of MCM5 and, especially, overexpression of R732A/K734A mutant MCM5 in the presence of R273H mutp53 inhibited mutp53-induced intracellular cGAMP accumulation under the same conditions (Supplementary Fig. 3o). Collectively, these results demonstrate that GOF mutp53 promotes cytoplasmic DNA and cGAMP accumulation, which is suppressed by MCM5 overexpression.

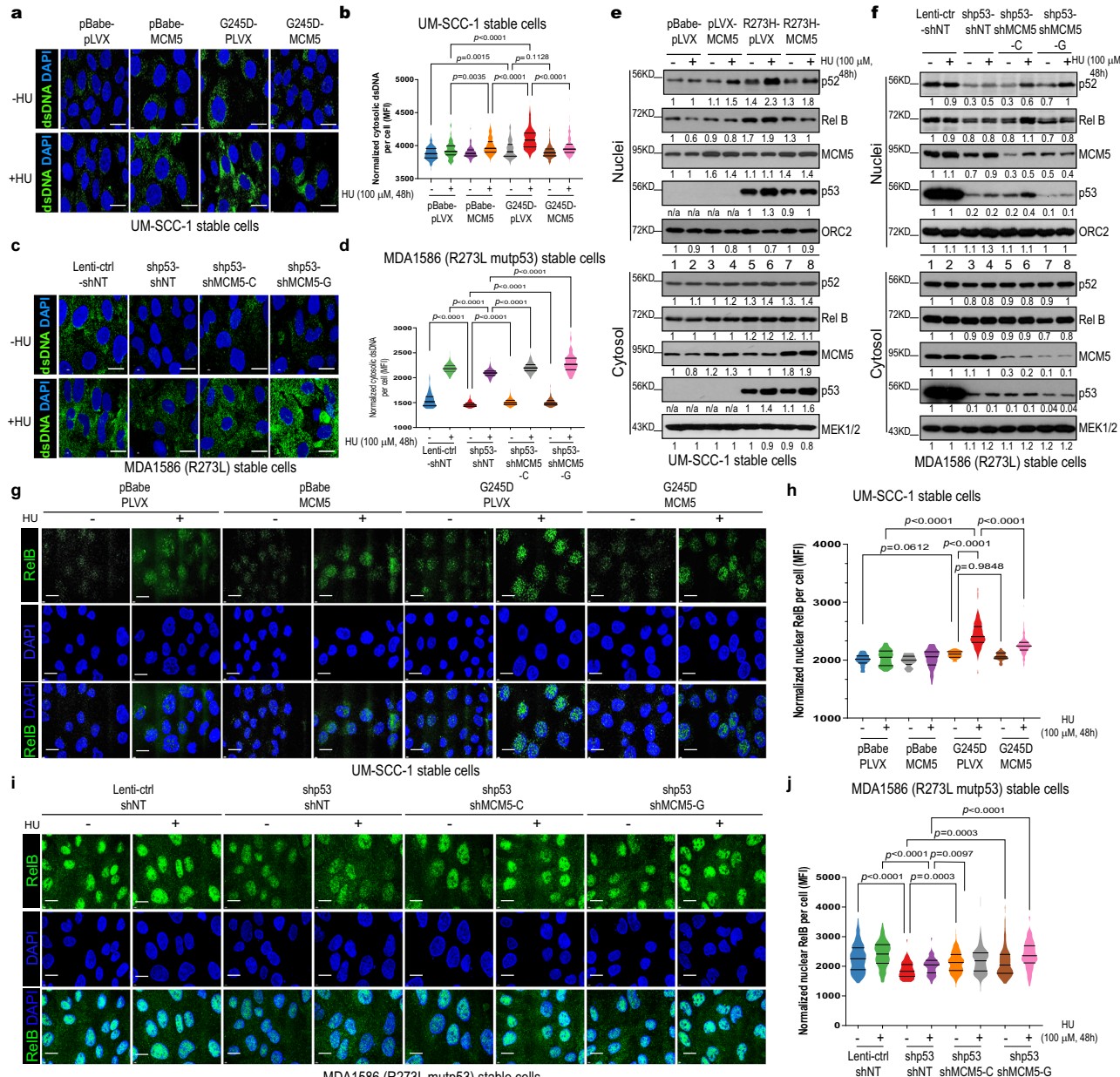

**Fig. 3 | GOF mutp53-MCM5-mediated replication stress and CIN stimulate cytosolic DNA accumulation and NC-NF-κB activation. a, c** Representative immunofluorescence (IF) staining images of cytosolic dsDNA in the indicated stable cell lines in the absence or presence of hydroxyurea (HU) (100 μM, 48 h). Scale bar, 10 μm. **b, d** Violin plots of the mean fluorescent intensity (MFI) per cell of cytosolic dsDNA in the indicated stable cell lines. Bars represent the median ± quartiles; $n = 167$ (pBabe-PLVX), 82 (pBabe-PLVX + HU), 110 (pBabe-MCM5), 227 (pBabe-MCM5 + HU), 106 (G245D-PLVX), 227 (G245D-PLVX + HU), 221 (G245D-MCM5), 183 (G245D-MCM5 + HU) **(b)**; $n = 560$ (Lenti-ctrl-shNT), 202 (Lenti-ctrl-shNT + HU), 346 (shp53-shNT), 196 (shp53-shNT + HU), 376 (shp53-shMCM5-C), 259 (shp53-shMCM5-C + HU), 280 (shp53-shMCM5-G), 98 (shp53-shMCM5-G + HU) **(d)**. **e, f** Western blot analyses of the cytosolic and nuclear fractions of the indicated

cells. **g, i** Representative IF staining images of RelB in the indicated stable cell lines in the absence or presence of hydroxyurea (HU; 100 μM, 48 h). Scale bar, 10 μm. **h, j** Violin plots of the MFI of nuclear RelB per cell in the indicated stable cell lines. Bars represent the median ± quartiles; $n = 48$ (pBabe-pLVX), 54 (pBabe-pLVX + HU), 26 (pBabe-MCM5), 78 (pBabe-MCM5 + HU), 41 (G245D-pLVX), 74 (G245D-pLVX + HU), 28 (G245D-MCM5), 105 (G245D-MCM5 + HU) **(h)**; $n = 250$ (Lenti-ctrl-shNT), 138 (Lenti-ctrl-shNT + HU), 73 (shp53-shNT), 105 (shp53-shNT + HU), 181 (shp53-shMCM5-C), 203 (shp53-shMCM5-C + HU), 174 (shp53-shMCM5-G), 112 (shp53-shMCM5-G + HU) **(j)**. Cells were incubated with doxycycline (200 ng/mL) for 48 h before further HU treatment **(c)**, **(d)**, **(f)**, **(i)**, and **(j)**. Significances were tested by the Kruskal–Wallis test with Dunn's multiple comparisons. Source data are provided as a Source Data file.

## GOF mutp53-MCM5-CIN-induced cytosolic DNA activates NC-NF-κB signaling

Cytosolic DNA activates the cGAS-STING signaling pathway through cGAMP, which in turn activates downstream TANK-binding kinase 1 (TBK1), promoting the phosphorylation and nuclear translocation of transcription factors such as IFN regulatory factor 3 (IRF3) and NF-κB[53,54]. To our surprise, mutp53 expression in UM-SCC-1 cells appeared to stimulate only NC-NF-κB signaling (i.e., a higher level of nuclear p52/RelB accumulation) when compared with control cells treated with or without hydroxyurea (Fig. 3e, lanes 5 vs. 1, 6 vs. 2; Fig. 3g, h; Supplementary Fig. 3p, lanes 7 vs. 5, 8 vs. 6 and Supplementary Fig. 3q, lanes 5 vs. 1, 6 vs. 2), whereas nuclear pIRF3 and canonical NF-κB signaling (i.e., nuclear p-p65) were largely unaffected (Supplementary Fig. 3p). Consistent with this, knockdown of endogenous mutp53 in MDA1586 or PCI-15B cells significantly decreased nuclear p52/RelB accumulation (Fig. 3f, lanes 3 vs. 1, 4 vs. 2; Fig. 3i, j; Supplementary Fig. 3r, lanes 7 vs. 5, 8 vs. 6 and Supplementary Fig. 3s, lanes 3 vs. 1, 4 vs. 2), suggesting that GOF mutp53 promotes NC-NF-κB signaling in these cells. In addition, knockdown of mutp53 in MDA1586 and PCI-15B cells also reduced the nuclear accumulation of p-p65 and/or pIRF3, especially in the presence of hydroxyurea (Supplementary Fig. 3r, lanes 8 vs. 6 and Supplementary Fig. 3s, lanes 3 vs. 1, 4 vs. 2), indicating mutp53 activated canonical NF-κB signaling in these cells. Nevertheless, our results showed that in the absence of hydroxyurea, NC-NF-κB signaling was activated by mutp53 in all 3 tested cell lines (UM-SCC-1, MDA1586, and PCI-15B) (Fig. 3e–j and Supplementary Fig. 3p–s), supporting previous observations that cGAS-STING signaling results in TBK1-dependent activation of p65/p50 canonical NF-κB signaling and also activates RelB/p52 NC-NF-κB signaling in a TBK1-independent manner[54]. Furthermore, overexpression of MCM5 and especially overexpression of R732A/K734A mutant MCM5 in UM-SCC-1 cells inhibited mutp53-induced nuclear RelB/p52 accumulation in response to hydroxyurea treatment (Fig. 3e, lanes 7 vs. 5, 8 vs. 6; Fig. 3g, h and Supplementary Fig. 3q, compare lanes 10 vs. 6 to 8 vs. 6), whereas further knockdown of MCM5 in mutp53-downregulated MDA1586 and PCI-15B cells restored the nuclear p52/RelB accumulation that had been impaired by mutp53 loss (Fig. 3f, lanes 6 vs. 4, 8 vs. 4; Fig. 3i, j and Supplementary Fig. 3s, lanes 5 vs. 3, 6 vs. 4, 7 vs. 3, 8 vs. 4). Taken together, our results strongly suggest that GOF mutp53-MCM5 signaling not only predisposes cells to replication stress and CIN that lead to cytosolic DNA and cGAMP accumulation, but also drives activation of NC-NF-κB signaling.

## GOF mutp53 promotes tumor cell invasion and metastasis through MCM5-CIN-cytosolic DNA-cGAS-STING-induced NC-NF-κB signaling

We and others have previously shown that the promotion of tumor cell invasion and metastasis is one of the most important features of mutp53 GOF[23,24,55]. In support of the involvement of cGAS-STING-signaling in NC-NF-κB activation, knockdown of either *CGAS* or *STING1* inhibited mutp53-mediated NC-NF-κB signaling (Fig. 4a–c). Moreover, knockdown of *CGAS, STING1,* or *RelB* in mutp53-overexpressing human UM-SCC-1 or mouse p53-null 4T1 stable cells (Fig. 4d–f) impaired not only mutp53-mediated in vitro transwell migration and invasion (Fig. 4g–p) but also mutp53-induced in vivo lung metastases after cells were injected into the tail veins of nude mice (Fig. 4q–s and Supplementary Fig. 4a–c) or the mammary fat pads of BALB/c mice (Fig. 4t–w and Supplementary Fig. 4d). Moreover, overexpression of MCM5 or R732A/K734A mutant MCM5 in UM-SCC-1 stable cells in the presence of mutp53 inhibited mutp53-induced in vitro transwell invasion (Supplementary Fig. 4e–h) and in vivo lung metastasis from cells injected into the tail veins of nude mice (Supplementary Fig. 4i, j). Consistently, while knockdown of R273L mutp53 in MDA1586 cells inhibited in vitro invasion and in vivo lung metastasis from tail-vein injection, further

knockdown of MCM5 in these cells rescued the impaired invasion and lung metastasis caused by mutp53 knockdown (Supplementary Fig. 4k–o). Taken together, these results demonstrate that MCM5-cGAS-STING-NC-NF-κB signaling plays an important role in mutp53 GOF-mediated tumor cell invasion and metastasis.

## GOF mutp53-MCM5-CIN-cytosolic DNA-cGAS-STING-induced NC-NF-κB signaling promotes tumor immunosuppression

Our results also showed that compared with the control, expression of murine R270H mutp53 in mouse breast cancer 4T1 cells exhibited GOF activity to promote tumor growth when cells were injected into the mammary fat pads of syngeneic BALB/c mice, but this GOF activity was abolished by further knockdown of *RelB* expression (Fig. 5a). In contrast, when the same group of 4T1 stable cells was orthotopically injected into immunodeficient SCID mice, all tumors grew faster than in immunocompetent BALB/c mice (Fig. 5b vs. 5a), but no mutp53 GOF activity was observed; neither expression of R270H mutp53 nor further knockdown of *RelB* had an impact on tumor growth when compared with the control in SCID mice (Fig. 5b). All these results indicate that R270H mutp53 promotes RelB-dependent tumor immunosuppression in immunocompetent BALB/c mice. In support of this, increased nuclear RelB localization was observed in R270H mutp53-expressing 4T1 tumors from BALB/c mice (Fig. 5c, d and Supplementary Fig. 5a). In addition, multiplex immunofluorescence staining or multiplexed ion beam imaging (MIBI) (Supplementary Fig. 5b, c) showed that R270H mutp53-expressing 4T1 tumors had less infiltration of CD3[+]CD8[+] and CD3[+] T cells, granzyme B[+] cells, dendritic cells (DCs) and lymphoid tissue-resident CD11b[+] classical dendritic cells (cDCs) in BALB/c mice compared with the control (Fig. 5e–m), suggesting that expression of R270H mutp53 engendered a immunosuppressive tumor microenvironment (TME) in 4T1 tumors. Moreover, further knockdown of RelB expression tended to restore the decreased immune cell infiltration induced by mutp53 expression (Fig. 5f, g, i, k, and m). This RelB-dependent immunosuppressive TME induced by R270H mutp53 strongly suggests that tumor cell-intrinsic mutp53-NC-NF-κB signaling plays an important role in mutp53-induced tumor immunosuppression.

## GOF mutp53-MCM5-CIN-cytosolic DNA-cGAS-STING-induced NC-NF-κB signaling antagonizes interferon (IFN) signaling

In support of our identification of mutp53-MCM5-cGAS-STING signaling, results of RNA-sequencing (RNAseq) and Gene Set Enrichment Analysis (GSEA) showed that IFN signaling (hallmarks of IFNα/γ response) and inflammation-related signaling (i.e., hallmarks of TNFα signaling via NF-κB + IL-6-JAK-STAT3 signaling + inflammatory response) pathways were among the signaling pathways that were most significantly altered by modulating expression of R273H mutp53, STING, RelB, and MCM5 in UM-SCC-1 and MDA1586 cells (Fig. 6a and Supplementary Fig. 5e). Specifically, expression of R273H mutp53 generally increased IFN pathway gene expression, but subsequent *STING1* knockdown reduced it (Fig. 6a–c and Supplementary Fig. 6a). However, in the presence of hydroxyurea, many R273H mutp53-induced IFN signaling genes were suppressed when compared with controls without hydroxyurea treatment (Fig. 6a and Supplementary Fig. 6b). More importantly, inhibition of NC-NF-κB signaling through *RelB* knockdown in p53-null UM-SCC-1 cells (Supplementary Fig. 5d) or R273H mutp53-expressing UM-SCC-1 cells resulted in increased expression of many IFN pathway genes (Fig. 6a–c; Supplementary Figs. 5e and 6a, c), suggesting that mutp53-induced NC-NF-κB signaling plays an important role in suppression of IFN signaling gene expression.

Our results also showed that MCM5 overexpression increases IFN pathway gene expression in the absence of mutp53 in UM-SCC-1 cells (Fig. 6a and Supplementary Fig. 6d–g). However, MCM5 overexpression in the presence of mutp53 reduced the expression of many

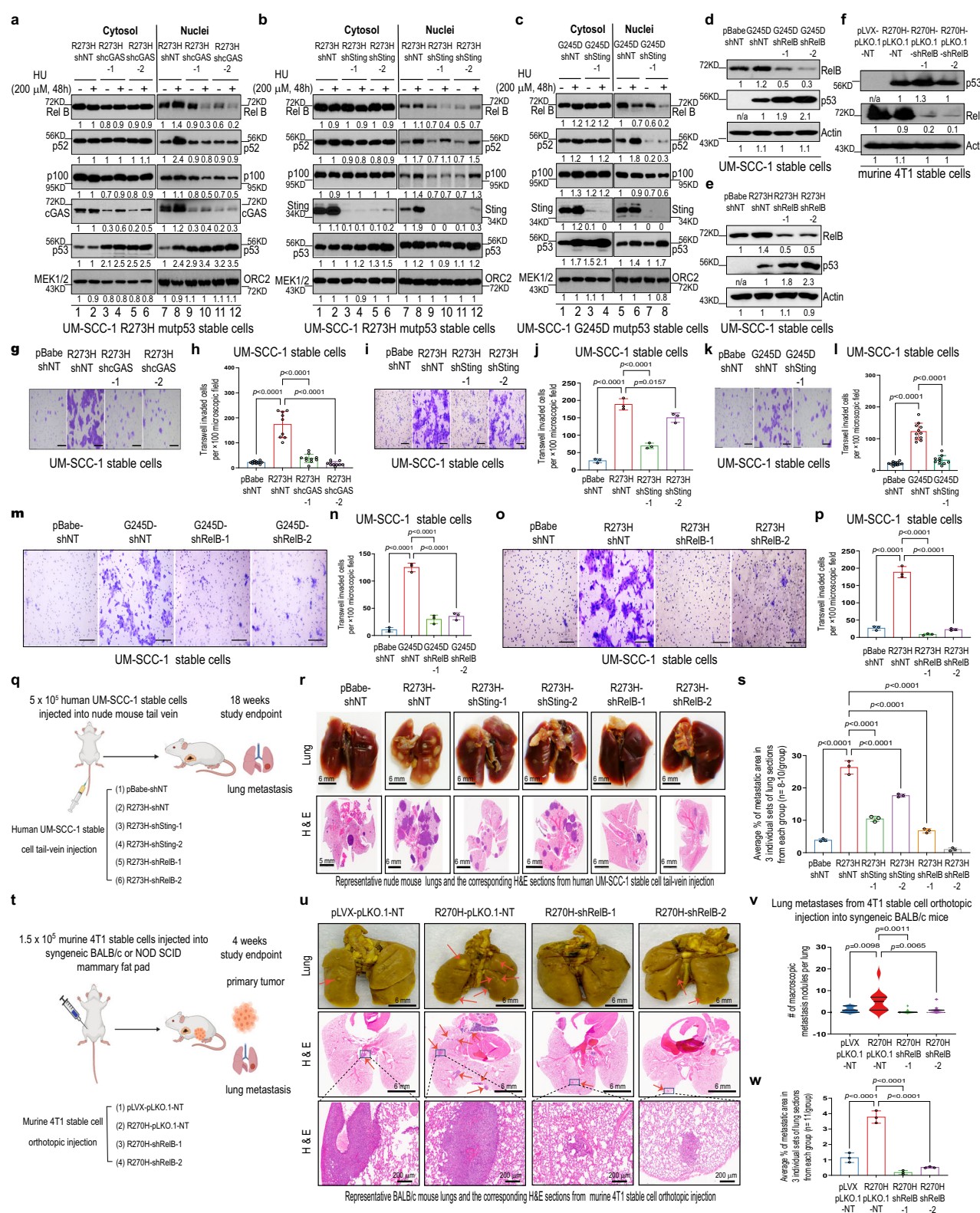

mutp53-induced IFN genes (Fig. 6a and Supplementary Fig. 6d, e, h–k). Consistently, knockdown of R273L mutp53 in MDA1586 cells reduced IFN signaling gene expression, but subsequent knockdown of MCM5 in these cells upregulated expression of many IFN pathway genes that had been lost due to mutp53 knockdown (Fig. 6a and Supplementary Fig. 6l–q). These data provide further evidence that MCM5 is involved in mutp53-mediated IFN signaling.

## GOF mutp53-MCM5-CIN-cytosolic DNA-cGAS-STING-NC-NF-κB-mediated IFN signaling inhibition and inflammation-related gene expression are associated with immunosuppression and tumor progression

While NC-NF-κB signaling was shown to inhibit IFN signaling, our results also indicated that knockdown of *RelB* impaired R273H mutp53-induced inflammation-related signaling pathways (Fig. 6a and

**Fig. 4 | GOF mutp53 promotes tumor cell invasion and metastasis through cGAS-STING-NC-NF-κB signaling. a–e** Western blot analyses of UM-SCC-1 stable cells with *CGAS* (**a**), *STING1* (**b**, **c**) or *RelB* knockdown (**d**, **e**). **f** Western blot analysis of murine 4T1 stable cells that were first stably introduced with mouse R270H mutp53 (equivalent to human R273H mutp53) and then with *RelB* knockdown. **g–p** *CGAS*, *STING* or *RelB* knockdown impairs mutp53-mediated migration and invasion. Shown are representative images (**g**, **i**, **k**, **m**, and **o**) and the summary graphs (**h**, **j**, **l**, **n**, and **p**) of the invasion of the indicated stable cell lines. *n* = 12 (pBabe-shNT), 9 (R273H-shNT), 10 (R273H-shcGAS-1), 10 (R273H-shcGAS-2) (**h**); *n* = 3 in each group (**j**, **n**, and **p**); *n* = 12 (pBabe-shNT), 11 (G245D-shNT), 10 (G245D-shSting-1) (**l**). Scale bar, 100 μm (**g**, **i**, **k**, **m**, and **o**). **q** and **t** Schematic representation created with BioRender.com of tail-vein injection of human UM-SCC-1 stable cell lines into nude mice (**q**) or orthotopic injection of murine 4T1 stable cell lines into BALB/c mice (**t**). **r**, **u** Representative macroscopic and corresponding microscopic hematoxylin and eosin (H & E) staining images of mouse lungs from nude mice 18 weeks after tail-vein injection with human UM-SCC-1 stable cells ($10^6$ cells/mouse) (**r**) or from BALB/c mice 4 weeks after mammary fat pad injection with the indicated mouse 4T1 stable cell lines ($1.5 \times 10^5$ cells/mouse) (**u**). Arrows: lung metastatic nodules/lesions. The lower panels in **u** are the magnified images of the corresponding box areas in the middle panels. **s** Mean percentage of lung microscopic tumor metastasis areas from **r**. *n* = 3 in each group. See the Methods for details. **v** Numbers of lung macroscopic metastasis nodules from **u**. *n* = 11 in each group. **w** Mean percentage of microscopic metastasis areas from **u**. *n* = 3 in each group. See the Methods for details. pBabe and pLVX: empty control vectors. shNT and pLKO.1-NT: non-target shRNA controls. Data shown present the mean ± SD (**h**), (**j**), (**l**), (**n**), (**p**), (**s**), and (**w**) or ± quartiles (**v**). Significances were tested using one-way ANOVA with Tukey's multiple comparisons test. Source data are provided as a Source Data file.

Supplementary Fig. 6r–z), suggesting that some genes in these pathways are positively regulated by NC-NF-κB signaling. Consistent with this, hierarchical clustering of single-sample GSEA (ssGSEA) for IFN and inflammation-related signaling pathways in HPV-negative and *TP53*-mutant oral squamous cell carcinoma (OSCC) in The Cancer Genome Atlas (TCGA) database identified a group of patients with low scores for the IFN pathway but high scores for the inflammation-related pathways (designated as "IFN-low/Inf-high" in Fig. 6d). This could indicate repression of IFN genes but activation of inflammation-related genes by NC-NF-κB signaling, as suggested by our cell line data. Importantly, the IFN-low/Inf-high patients had the worst overall, disease-specific, and progression-free survival among all the other groups (IFN-high/Inf-high [All High], IFN-low/Inf-low [All Low] and IFN-high/Inf-low) (Fig. 6e, f and Supplementary Fig. 7a). Consistent with this, compared with all other groups, IFN-low/Inf-high tumors exhibited a trend toward the strongest epithelial-mesenchymal transition (EMT) phenotype (Fig. 6g), which is often associated with HNSCC metastasis and adverse pathological features[56]. Moreover, although both IFN-low/Inf-high and All High tumors displayed more leukocytes and Th1 cell and lymphocyte infiltration fractions, indicating more immune activity than in other groups (Supplementary Fig. 7b–d), IFN-low/Inf-high tumors had the lowest infiltration of activated natural killer (NK) cells and T follicular helper (Tfh) cells among all groups (Fig. 6h, i), and had lower CD8+ T cells, IFN-γ response, and M1 macrophages—but more M0 macrophages—than did the IFN-high/Inf-low and All High groups (Fig. 6j and Supplementary Fig. 7e–g). In addition, our analyses indicated the presence of similar molecular patterns in *TP53* mutant larynx and lung squamous carcinomas, in which a trend toward high EMT signature and low infiltration of activated NK, Tfh, and CD8+ T cells was also observed in tumors with the "IFN-low/Inf-high" molecular phenotype (Supplementary Fig. 8a–j). Together, these results showed that IFN-low/Inf-high tumors are associated with an immunosuppressive tumor microenvironment, strongly suggesting that mutp53-MCMs-CIN-cytosolic DNA-STING-induced NC-NF-κB signaling, through inhibition of IFN and activation of inflammation-related signaling in tumor cells, has an important role in tumor progression and the resistance to anti-tumor immunity of tumors with *TP53* mutations (Fig. 6k).

## Discussion

p53 was characterized as the "guardian of the genome" over 30 years ago[5]. Its importance in maintaining genomic integrity and inhibiting tumor development was recently re-demonstrated by Lowe and colleagues in a study showing that p53 can enable the predictable development of CIN that is required for p53 loss-induced tumor malignancy[6]. However, whether the development of CIN is attributable to only p53 LOF or whether mutp53 GOF can also contribute to CIN development is still not clear. Since the concept of mutp53 GOF was introduced in the 1990s[57,58], many context-dependent and conflicting findings regarding the oncogenic GOF phenotypes associated with overexpression of mutp53 proteins and related mechanisms have been reported[25–33]. Of note, an important recent study from Pietenpol and colleagues used both in vitro studies and an immunocompromised nude mouse model to convincingly demonstrate that acquisition of CIN is also required for mutp53-associated GOF phenotypes[34]. This intriguing finding suggests that the acquisition of aneuploidy may be a unifying mechanism that accounts for many context-specific GOF phenotypes previously attributed to mutp53 proteins[34]. Since all inactivating p53 mutations will result in CIN, the different roles of LOF, dominant-negative, and GOF p53 mutations in enabling CIN development are still not clear, although an increased frequency of aneuploidy has been observed in p53 GOF mutant isogenic cell lines[34]. The work presented here confirms these other investigators' findings, highlights the importance of CIN in p53 mutation-induced oncogenic phenotypes, and further demonstrates that there is indeed a causal relationship between mutp53 GOF and CIN, in which mutp53 actively predisposes cells to replication stress and CIN through interaction with MCMs and consequent deregulation of the MCM2-7 complex. Taken in aggregate with these recent reports from other laboratories showing that CIN is the prerequisite for p53 inactivation-induced malignancy and for mutp53 GOF-associated phenotypes[6,34], our findings reveal the underlying mechanistic relationships between mutp53 GOF activity and enhanced propensity to CIN.

It has been shown previously that CIN can activate chronic cGAS-STING signaling, which can promote tumor metastasis and survival through TBK1-independent activation of NC-NF-κB and IL-6-STAT3 signaling[14,16,52,59]. In contrast, acute activation of the cGAS-STING pathway often leads to anti-tumor immunity through TBK1-dependent activation of IRF3 and canonical NF-κB p65/p50 signaling, which mediate the transcription of type I IFN signaling[53,60–62]. Given these dual but opposing roles (i.e., anti-tumorigenic vs. pro-tumorigenic roles) of the cGAS-STING pathway, understanding how this pathway is regulated in tumor cells to lead to different outcomes is very important for studies of tumorigenesis and tumor progression. Consistent with this, the cGAS-STING pathway is rarely inactivated in primary tumors, but many tumors develop a variety of still poorly understood mechanisms that suppress downstream type I IFN signaling under chronic cGAS-STING signaling triggered by CIN[52,59]. Here, in addition to supporting that CIN induces NC-NF-κB signaling[16,52], our results showed that mutp53-MCM5-CIN-cGAS-STING-induced NC-NF-κB signaling not only promotes metastasis but also has an inhibitory role in IFN signaling. Our findings are consistent with previous studies showing that activation of NC-NF-κB antagonizes canonical NF-κB signaling and STING-mediated type I IFN signaling[63,64]. Therefore, these results strongly support an important role for mutp53-CIN in inducing tumor cell-intrinsic resistance to anticancer immunity through NC-NF-κB-mediated inhibition of IFN signaling or IFN tachyphylaxis—a process of reduction in IFN responsiveness to repetitive stimulation—which was recently shown by Bakhoum and colleagues to play an important role in CIN-induced immune suppression and metastasis[65]. Moreover, NC-NF-κB signaling also positively regulates some genes

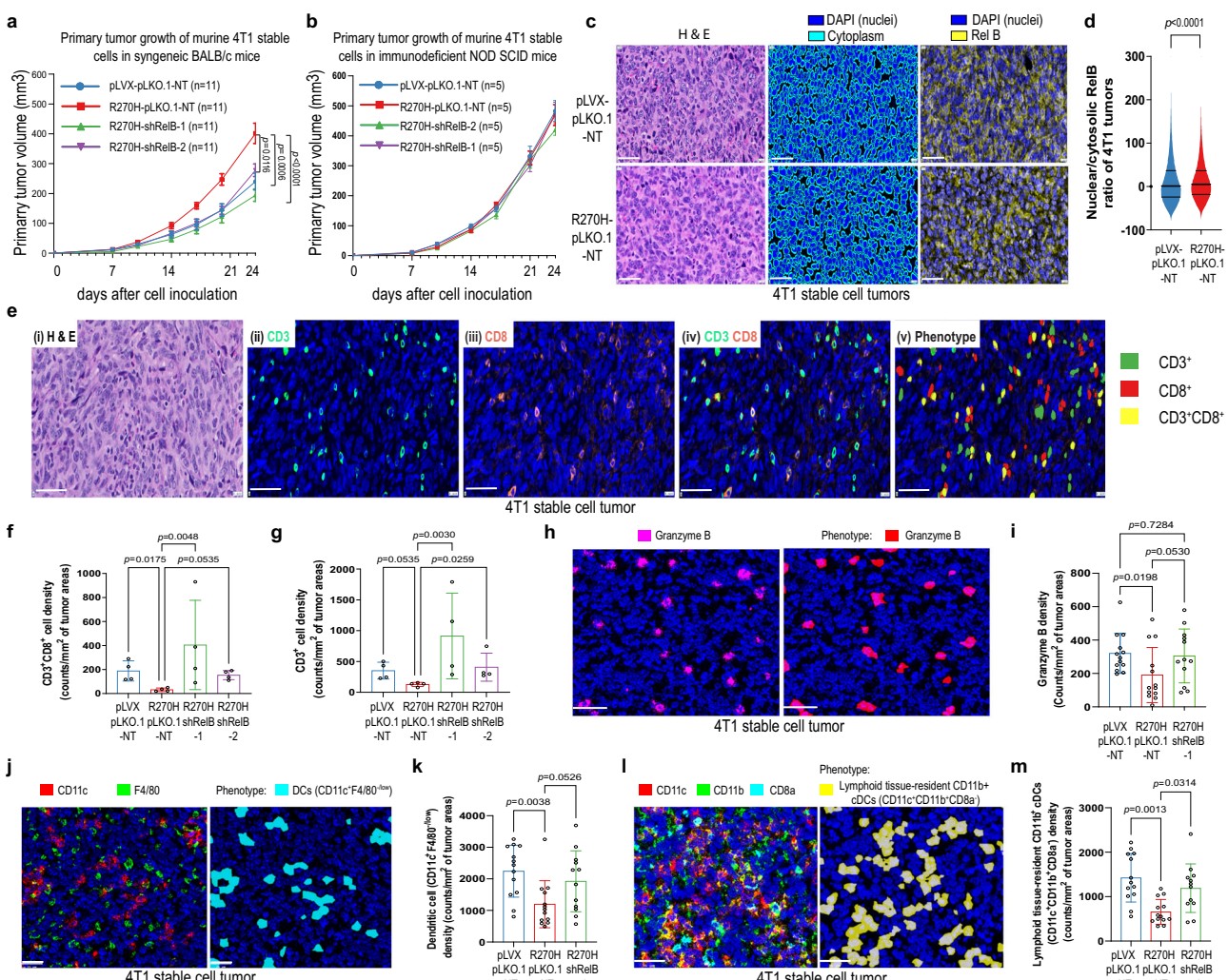

**Fig. 5 | GOF mutp53-NC-NF-κB signaling promotes tumor immunosuppression.**
**a**, **b** Primary tumor growth of mouse 4T1 stable cells orthotopically injected into BALB/c (**a**) or NOD SCID (**b**) mice. $n = 11$ in each group (**a**); $n = 5$ in each group (**b**). **c** Representative H & E and corresponding RelB immunofluorescent images in 4T1 stable cell tumors from BALB/c mice. Middle panel: cytosol vs. nuclei images from artificial intelligence (AI)-based analyses (see Methods). Scale bar, 50 μm. **d** Quantitative RelB nuclei/cytosol ratio from 4T1 stable cell tumors from BALB/c mice. $n = 57{,}370$ (pLVX-pLKO.1-NT), 78,073 (R270H-pLKO.1-NT). **e** Representative H & E and corresponding multiplex CD3 and CD8 immunofluorescent images from a 4T1 stable cell tumor from BALB/c mouse (panels i–iv). Panel v: AI-based CD3 and CD8 phenotyping image of panels ii–iv. Scale bar, 50 μm. **f**, **g** Quantitative results of CD3+ and CD8+ cells in tumor areas of 4T1 stable cell tumors from BALB/c mice (see Methods). $n = 4$ in each group. Each point represents the data generated from

counting T cells in all tumor areas ($0.3$–$1.7 \times 10^6$ cells) of a whole section from different tumors in each group. **h**, **j**, and **l** Representative MIBI images (*left*) and phenotypes (*right*) of immune cell markers as indicated in 4T1 stable cell tumors from BALB/c mice. Scale bars, 36 μm (**h**, **l**) and 26 μm (**j**). **i**, **k**, and **m** Quantitative results of granzyme B+ cells (**i**), DCs (**k**), and lymphoid tissue-resident CD11b+ cDCs (**m**) in 4T1 stable cell tumors from BALB/c mice. $n = 13$ (pLVX-pLKO.1-NT), 13 (R270H-pLKO.1-NT), 12 (R270H-shRelB1) (**i**, **k**, and **m**). Data shown present the mean ± SEM (**a**) and (**b**), or ±quartiles (**d**), or ±SD (**f**), (**g**), (**i**), (**k**), and (**m**). Significances were tested using one-way ANOVA with Tukey's multiple comparisons test (**a**), two-tailed Wilcoxon-signed rank test (**d**), Kruskal–Wallis test with Dunn's multiple comparisons test (**f**), (**g**), (**i**), (**k**), and (**m**). Source data are provided as a Source Data file.

---

involved in inflammation-related signaling, including IL-6-STAT3 signaling (Fig. 6a and Supplementary Fig. 6r–z), suggesting that it may play an important role in pro-tumorigenic inflammation. Given that the immune system, in addition to its anti-tumor immunity, is also involved in tumor-promoting inflammation[66] and selection of tumor suppressor inactivation[67] during tumor development and progression, our identification of GOF mutp53-induced, CIN-cGAS-STING–driven NC-NF-κB activation has provided us a novel mechanistic insight into how tumor cells with inactivating p53 mutations interact with immune system to evade anti-tumor immunity.

It has previously been shown that GOF mutp53 can activate canonical NF-κB signaling through direct interaction with p65 (RelA)[68-71] and DAB2IP[72] or NC-NF-κB signaling through stimulation of *NFKB2* (p52/p100 subunit) expression[73,74]. Here, our findings

provide additional evidence of the complexity of mutp53-mediated NF-κB activation. Since canonical NF-κB signaling is rapid and transient, whereas NC-NF-κB signaling is usually slow but persistent[75], our identification of mutp53-induced and MCM5-CIN-cGAS-STING-mediated activation of NC-NF-κB may have important implications for understanding the previously described GOF role of mutp53 in prolonged NF-κB activation and chronic inflammation[68]. Moreover, since canonical NF-κB activation requires stimulation by extracellular ligands, whereas NC-NF-κB signaling can be activated by CIN[16,52], our finding of GOF mutp53-MCM5-CIN-cGAS-STING-NC-NF-κB signaling not only demonstrates a novel GOF mechanism of mutp53 but also strongly suggests that NC-NF-κB activation is a common mechanism involved in all inactivating p53 mutations that lead to CIN.

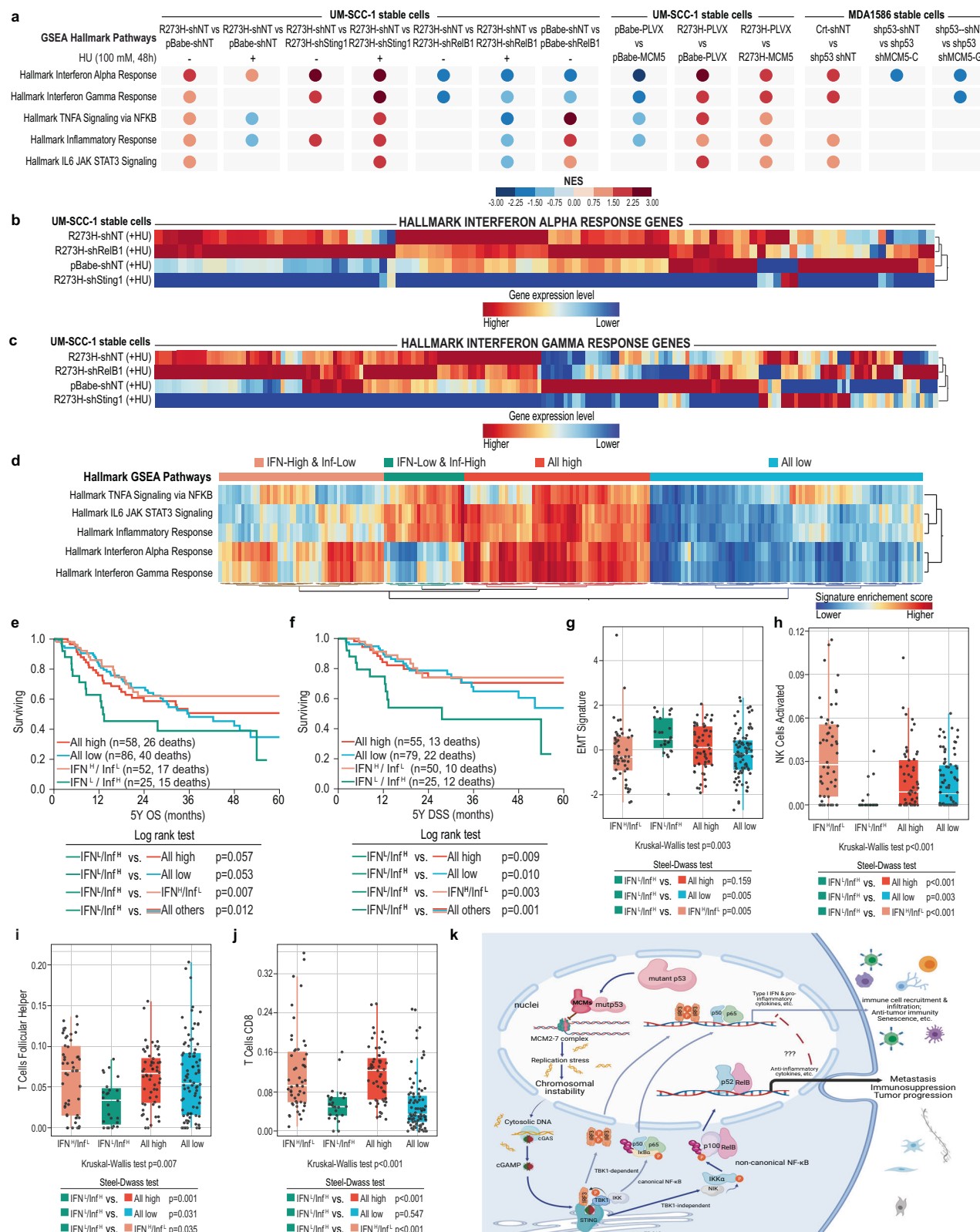

Although our current study focused on the MCM5-G245D or -R273H mutp53 interaction because MCM5 has a much higher R273H/L mutp53-binding affinity than do other MCM subunits, R273H mutp53 also interacts with other subunits of the MCM2-7 complex (Fig. 1f, g)[76]. Interestingly, whereas R248Q mutp53 did not bind to MCM5 well (Fig. 1d, lane 7 and Supplementary Fig. 1f, lane 6), it strongly bound to MCM7 (Supplementary Fig. 1g, lane 4). More importantly, R248Q

mutp53 showed a greater potential to induce replication stress than did R273H mutp53 in UM-SCC-1 cells even though its expression level was lower than that of R273H mutp53 (Supplementary Fig. 1h, lanes 5 and 6 vs. 3 and 4). Since the interaction with MCM7 and/or other MCM subunits, like the interaction with MCM5, is also likely to disrupt the stoichiometry of the MCM2-7 complex, especially when MCM2-7 complex assembly is much needed (e.g., under replication stress), it

**Fig. 6 | GOF mutp53-induced MCM5-STING-NC-NF-κB signaling antagonizes IFN signaling and regulates inflammation-related genes associated with immunosuppression and tumor progression of OSCC. a** Summary of normalized enrichment scores (NES) of GSEA Hallmark pathways of IFN and inflammation-related signaling that were significantly enriched (false-discovery rate q-values [FDRq] <25%]) for the indicated comparisons of cell lines. +HU: 100 μM, 48 h. **b**, **c** Hierarchical clustering analyses of the 97 genes (x-axis) in the Hallmark IFNα response gene set (**b**) and of the 200 genes in the Hallmark IFNγ response gene set (**c**) for the indicated comparisons of UM-SCC-1 stable cell lines in the presence of HU (+HU: 100 μM, 48 h). **d** Hierarchical clustering analysis of single-sample GSEA scores for the Hallmark GSEA IFN and inflammation-related (Inf) signaling pathways for each of the 221 TCGA patients with HPV-negative, TP53-mutant OSCC (x-axis). The analysis revealed 4 patient groups with distinct gene expression profiles. **e**, **f** Five-year overall survival (OS) and disease-specific survival (DSS) for the 221

patients with HPV-negative TP53-mutant OSCC with the indicated IFN and Inf gene expression profiles. L, low; H, high. **g–j** Box plots of EMT and immune enrichment scores of the 221 HPV-negative and TP53-mutant OSCCs with the indicated IFN and Inf gene expression profiles. The number of patients in each group was IFN$^H$/Inf$^L$ = 52, IFN$^L$/Inf$^H$ = 25, All High = 58, and All Low = 86. Shown are the mean immune enrichment scores (±SD) in each group. Boxes represent the interquartile range (IQR) and the horizontal line indicates the median. The whiskers extend to the last data point within 1.5×IQR. Significances were tested using Kruskal–Wallis test and the two-sided steel-dwass test (**g–j**). **k** A working model of the mutp53 GOF mechanism involving mutp53-MCMs-CIN-cytosolic DNA-cGAS-STING-NC-NF-κB signaling that promotes tumor metastasis, immunosuppression, and tumor progression. Created with BioRender.com (**k**). Source data are provided as a Source Data file.

strongly suggests that GOF mutp53s, through their differential impacts on the different subunits of the MCM2-7 complex, predispose cells to DNA replication stress and genomic instability that lead to mutp53 GOF activities.

In summary, our work reveals a tumor cell-intrinsic mechanism of mutp53 GOF involving mutp53-MCMs-induced CIN, leading to the activation of cytosolic DNA-cGAS-STING-NC-NF-κB signaling, which plays an important role in promoting metastasis and immunosuppression (Fig. 6k). Supported by recent studies showing that the acquisition of aneuploidy drives mutp53-associated GOF phenotypes[34], that STING activity is often increased in TP53−mutant compared with wt TP53 tumors[77], and that cGAS and STING are required for CIN-driven tumor progression[65], our findings provide significant insight into not only the GOF mechanisms of TP53 mutations but also all inactivating mutations of p53 that lead to genomic instability to promote immune suppression and metastasis during tumor development and tumor progression. This suggests that a strategy for targeting tumor cell-intrinsic NC-NF-κB signaling could be an impactful approach to treating cancers with TP53 inactivation-induced CIN.

## Methods
### Cell lines and cell culture
Immortalized human normal head and neck epithelial hTERT HAK cl41 cells[78] (wtp53) (provided by Dr. Aloysius J. Klingelhutz, University of Iowa Research Foundation) were cultured at 37 °C in a 5% CO$_2$ atmosphere in oral keratinocyte medium (ScienCell, #2611) supplemented with oral keratinocyte growth supplement (ScienCell, #2652). UM-SCC-1 (p53 protein-deficient, UM-SCC-1 cell has splice site mutation rendering the cells null for p53 protein expression and resulting in loss of p53 function[79]) (provided by Dr. Thomas E. Carey, University of Michigan), MDA1586 (R273L mutp53) (provided by Dr. Peter Sacks, New York University), PCI-15B (R273C mutp53) (provided by Dr. Jennifer Grandis, University of Pittsburgh School of Medicine), Detroit 562 (R175H mutp53) (ATCC, CCL-138), HN5 (C238S mutp53) (provided by Dr. D. M. Easty, Ludwig Institute for Cancer Research), Ca9-22 (R248W mutp53) (Japan Health Science Research Resource Bank), 4T1 (p53-null) (MD Anderson Cytogenetics and Cell Authentication Core), HEK293-FT (Thermofisher, R70007), and all stable cell lines subsequently established from these cells were cultured at 37 °C under 5% CO$_2$ in Dulbecco's modified Eagle medium (DMEM) supplemented with 10% fetal calf serum and 2 mM L-glutamine in the presence of penicillin (50 U/mL) and streptomycin (50 μg/mL). All the cell lines used in this study tested negative for mycoplasma and were authenticated and verified by short tandem repeat profiling performed on cellular DNA submitted to the MD Anderson Cytogenetics and Cell Authentication Core Facility or the Fragment Analysis Facility of Johns Hopkins University. The detailed short tandem repeats allelic patterns of each cell line were reported previously[80] and are listed in Supplementary Data 2.

### Overexpression and knockdown DNA constructs
Retroviral pBabe-puro-based human wt/mutp53 overexpression and lentiviral pLVHMshp53 vectors were described previously[81,82]. wt/mutp53s were synthesized by polymerase chain reaction (PCR) and cloned into an XbaI-digested Flag-HA-pcDNA3.1 vector using the In-Fusion cloning strategy (Takara Bio) to get Flag and HA double-tagged constructs. Similarly, MCMs were synthesized by reverse transcription PCR and cloned into EcoRI-digested pcDNA3.1/V5-His-A or pLVX-IRES-mCherry vectors using In-Fusion cloning. A TP53 CRISPR/Cas9 knock-out plasmid was obtained from Santa Cruz Biotech. Murine R270H mutp53 was generated by site-directed mutagenesis using pMXs-p53 as a template and then cloned into a pLVX-IRES-hygro vector using In-Fusion cloning. The sequences of all the constructs were validated by sequencing analyses. Lentiviral short hairpin RNAs (shRNAs) against MCM5, CGAS, STING1 (TMEM173), and human and murine RelB were purchased from Dharmacon or Sigma-Aldrich. Information on all the vectors, primers, and shRNAs is listed in Supplementary Data 3.

### Generation of stable cell lines
hTERK HAK Cl41-p53KO cells were obtained from hTERT HAK Cl41 cells by transient transfection (using Lipofectamine 2000) with the TP53 CRISPR/Cas9 knockout plasmid and then sorting by flow cytometry for single-cell GFP-positive clones. Stable hTERK HAK Cl41-p53KO-c1 and UM-SCC-1 cells overexpressing wtp53 or various mutp53s were obtained from pools of cells after retroviral infection and puromycin (1 μg/mL) selection. Retroviruses used for infections were generated by transfecting various pBabe wtp53 or mutp53 vectors with the packaging vectors pCMV-VSV-G and pCMV-gag-pol into HEK293-FT cells. MCM5 or MCM5 R732A/K734A expression was achieved by lentiviral infection using virus generated from HEK293-FT cells transfected with a pLVX-MCM5/MCM5 R732A & K734A-IRES-mCherry plasmid and the lentiviral packaging vectors pCMV-dR8.2 dvpr and pMD2.G; cells stably expressing MCM5 or MCM5 R732A/K734A were selected and sorted for mCherry expression. Doxycycline-inducible MCM5 or MCM2 expression was achieved by lentiviral infection using a virus generated from HEK293-FT cells transfected with TRE3G-MCM5/MCM2-PGK-Tet3G-bsd and the lentiviral packaging vectors, followed by blasticidin (ThermoFisher, #A1113903) selection. Murine R270H mutp53 expression was achieved by lentiviral infection using a virus generated from HEK293-FT cells transfected with a pLVX-R270H-IRES-hygro plasmid and its packaging vectors psPAX2 and pMD2.G, followed by hygromycin (ThermoFisher, #10687010) selection. Stable knockdown of TP53 was achieved by lentiviral infection using a virus generated from HEK293-FT cells transfected with pLVHMshp53 and the lentiviral packaging vectors, followed by sorting of GFP-positive cells. Stable knockdown of cGAS, STING1, and RELB was achieved by lentiviral infection using a virus generated from HEK293-FT cells transfected with pGIPZ-shRNAs and the lentiviral packaging vectors, followed by sorting of GFP-positive cells. Stable knockdown of MCM5 was achieved by lentiviral infection using a virus generated from

HEK293-FT cells transfected with doxycycline-inducible SMART-shRNAs-mCMV-TurboRFP vector and the lentiviral packaging vectors, followed by sorting of RFP-positive cells in the presence of doxycycline (200 ng/mL) (Fisher Scientific #BP26531). Stable knockdown of murine *RELB* was achieved using a virus generated from HEK293-FT cells transfected with pLKO.1-puro-shRNAs and the lentiviral packaging vectors, followed by puromycin selection. Information on all the plasmids used is provided in Supplementary Data 3.

## Mutp53 interactome screening by quantitative immunoprecipitation using stable isotope labeling with amino acids in cell culture (SILAC)

To identify GOF mutp53 interactome, we selected p53-deficient UM-SCC-1 cell line and G245D mutp53 since our previous study showed that overexpression of G245D mutp53 alone or in combination with expression of a constitutively active phosphoinositide 3-kinase PIK3CA H1047R in UM-SCC-1 cells exhibited a great mutp53 GOF function to promote cell invasive growth and metastasis in an orthotopic nude mouse model (manuscript in preparation). To this end, UM-SCC-1-pBabe, UM-SCC-1-mutp53G245D, and UM-SCC-1-mutp53G245D & PIK3CA H1047R cells were labeled via passaging for up to 8 cell divisions in DMEM containing L-arginine (Arg 0) and L-lysine (Lys 0) ("light") or L-arginine-U-$^{13}C_6$$^{15}N_4$ (Arg 10) and L-lysine-U-$^{13}C_6$$^{15}N_2$ (Lys 8) ("heavy"). For immunoprecipitation (IP)-mass spectrometry (MS), $1 \times 10^7$ cells were lysed with NETN buffer (0.5% NP-40, 1 mM EDTA, 20 mM Tris-HCl [pH 8.0], and 100 mM NaCl with freshly added protease and phosphatase inhibitors (1 mM phenylmethylsulfonyl fluoride [PMSF], 1 µg/mL pepstatin, 1 µg/mL aprotinin, 1 mM sodium vanadate, 15 mM β-glycerophosphate, and 50 mM NaF) on ice for 30 min. The lysates were incubated with 20 µL of p53 antibody-agarose (see Supplementary Data 4) for 2 h at 4 °C. After 4 washes with NETN buffer, the paired samples were combined and then boiled in 2× Laemmli buffer (4% sodium dodecyl sulfate [SDS], 20% glycerol, 120 mM Tris-HCl, pH 6.8) before SDS-polyacrylamide gel electrophoresis (SDS-PAGE). The entire sample lane of SDS gel was excised into 3 fractions and then destained, and digested with trypsin at 37 °C overnight. On the next day, the digested peptides were extracted with acetonitrile and vacuum-dried before MS analyses.

## Mass spectrometry and data analysis

The digested peptides were dissolved in 0.1% formic acid, then loaded into a nanoscale liquid chromatography system (EASY-nLC 1000 liquid chromatography system, ThermoFisher), and separated with a 75-min discontinuous gradient of 4–24% ACN/0.1% formic acid. The flow rate was set at 800 nL/min. After peptide separation, they were analyzed by Q Exactive Plus (ThermoFisher). The settings for mass spectrometry included data-dependent mode, precursor MS scan range at 375–1300 m/z; mass resolution for MS1 and MS2 of 140,000 and 17,500, respectively; AGC for MS1 and MS2 of $3 \times 10^6$ and $2 \times 10^5$, respectively; the top 25 precursor ions were selected to analyze by MS2 with collision energy at 27.

Raw MS data were processed by MaxQuant software (version 1.5.8.3) with an FDR < 0.01 at the PSM and protein level. MS tolerance for MS1 was 20 ppm, and for MS2 was 0.5 Da. Oxidation of methionine, acetylation of N-term, and carbamidomethyl of cysteine were selected as variable modifications. The minimum length for peptides was 7 amino acids. A target-decoy approach with a reversed database was used for protein and peptide identification. The human FASTA database was downloaded from UniProt (April 2017) containing 70946 entries. Peptides and proteins were quantified by MaxQuant with default settings. Two different cell line comparisons (i.e., G245D mutp53 vs. pBabe; G245D mutp53 and PIK3CA H1047R vs. pBabe) were analyzed. In each group, duplicates with both heavy (H)/light (L) and reverse L/H SILAC labeling were used for IP with 2 different p53 antibodies (DO-1 and Pab240, respectively; see Supplementary Data 4).

Therefore, in total, 8 experiments were carried out. MS data were searched in the Maxquant database. The cutoff values for candidates were a median fold change higher than 1.5 (G245D/pBabe or G245D + PIK3CA/pBabe) and a *P*-value (two-sided *t*-test) lower than 0.05 in all 8 experiments. Finally, metascape enrichment analysis[47] was used to analyze the results.

## Immunoprecipitation

Cells were lysed in an extraction buffer (30 mM Tris-HCl, pH 7.4, 1% Triton-X-100, 150 mM NaCl, 10% glycerol, 10 mM MgCl$_2$, 10 mM KCl, 2 mM CaCl$_2$, 1 mM PMSF, 1 mM NaF, and 1× protease inhibitor cocktail [Millipore Sigma #I1697498001]). Lysates were clarified by centrifugation at $15,000 \times g$ for 10 min. For IP, the supernatants were incubated with the appropriate antibodies overnight at 4 °C with constant rotation. After the addition of Dynabeads protein G beads (ThermoFisher #10003D), the lysates were incubated for an additional 5 to 10 min at room temperature. Next, the beads were separated magnetically and washed up to 6 times with the extraction buffer before the addition of SDS Laemmli sample buffer for SDS-PAGE and immunoblotting. The antibodies used are listed in Supplementary Data 4.

## Immunoblotting

In addition to nuclear and chromatin extracts, total cell extracts were obtained from cells pelleted and lysed by radioimmunoprecipitation assay (RIPA) buffer with 1× protease inhibitor cocktail. The concentration of total cell extracts was determined using a Bradford protein assay. Proteins were separated by SDS-PAGE and transferred to polyvinylidene difluoride membranes. See Supplementary Data 4 for antibody information. Blots were developed with SuperSignal West Pico Chemiluminescent Substrate (ThermoFisher) according to the manufacturer's instructions, exposed to autoradiographic films (LabScientific by ThermoFischer), and scanned on an Epson Perfection V800 Photo Scanner. Protein bands' densitometry was measured and calculated using NIH Image J software. Uncropped and unprocessed scans of the blots are provided in the Source Data files.

## Nuclear extracts and chromatin fraction isolation

A previously described method[83] was used with modifications. Briefly, to isolate nuclear extracts, cells were resuspended in buffer A (10 mM HEPES, pH 7.9, 10 mM KCl, 1.5 mM MgCl$_2$, 0.34 M sucrose, 10% glycerol, 1 mM dithiothreitol [DTT], 1× protease inhibitor cocktail, 0.1 mM PMSF). Triton-X-100 (0.1%) was added, and the cells were incubated for 5 to 10 min on ice. Nuclei were collected as pellets (P1) by low-speed centrifugation (5 min, 1300 g, 4 °C). The supernatant (the cytosolic fraction) was further clarified by high-speed centrifugation (15 min, 15,000 g, 4 °C) to remove cell debris and insoluble aggregates. To obtain nuclear extracts, nuclear pellets were washed twice with buffer A, and then directly resuspended and dissolved in Laemmli buffer with sonication in a Branson digital sonicator. Alternatively, to isolate the chromatin fraction, nuclear pellets (P1) were washed once in buffer A and then lysed in buffer B (3 mM EDTA, 0.2 mM EGTA, 1 mM DTT, 1× protease inhibitor cocktail). The insoluble chromatin fraction was collected by centrifugation (5 min, 1700 g, 4 °C) and washed once in buffer B under the same conditions. The final insoluble chromatin pellet was resuspended and dissolved in Laemmli buffer with sonication. The concentrations of nuclear or chromatin proteins were determined by a Pierce 660-nm protein assay (#22660) with Ionic Detergent Compatibility Reagent (#22663) before the nuclear or chromatin fraction was subjected to SDS-PAGE and immunoblotting.

## Metaphase chromosome spread preparation

For hTERK HAK cl41-p53KO-c1 stable cells, $1.2 \times 10^5$ cells were plated in a 6-well plate with 3 mL oral keratinocyte medium, incubated for 24 h

at 37 °C, and then treated with hydroxyurea (100 μM) for 24 to 72 h at 37 °C. The cells were then washed once with phosphate-buffered saline (PBS) and treated with colcemid (0.2 mg/mL, Millipore Sigma #10295892001) for 18 h. For MDA1586 cells, $6 \times 10^4$ cells were plated in a 6-well plate with 3 mL DMEM for 24 h at 37 °C and then treated with hydroxyurea (200 μM) for 18 h at 37 °C. The cells were then washed once with PBS and treated with colcemid (0.15 mg/mL) for 6 h. After colcemid treatment, cells were trypsinized, transferred to 15-mL conical tubes, and centrifuged. The cell pellets were treated with hypotonic solution (0.075 M KCl) for 20 min at room temperature and then fixed in a methanol and acetic acid mixture (3:1 v/v) for 15 min at room temperature and washed 3 times with fixative. The samples were then sent to MD Anderson's Cytogenetics and Cell Authentication Core Facility for the generation of chromosome spreads on glass slides according to the standard protocol. Slides were stained with 4% Giemsa and analyzed for various chromosomal aberrations, including chromosome and chromatid breaks, fusions, fragments, and tetraploidy. At least 35 metaphases were analyzed from each sample. Images were captured using a Nikon 80i microscope equipped with karyotyping software from Applied Spectral Imaging, Inc.

### DNA fiber assay

Cells were treated with 25 mM CldU for 20 min followed by 200 μM hydroxyurea and 250 μM IdU for 4 h. Then, cells were lysed in spreading buffer (200 mM Tris-HCl pH 7.4, 50 mM EDTA, 0.5% SDS), streaked onto glass slides, dried, and fixed in 3:1 methanol/acetic acid for 10 min. DNA was denatured with 2 N HCl for 30 min, blocked in 10% horse serum, and stained overnight with rat anti-BrdU antibody and 1:300 mouse anti-BrdU antibody overnight at 4 °C. Slides were washed in TBST containing 1 M salt followed by PBS, and then stained with secondary antibodies (anti-rat Alexa Fluor 568 and anti-mouse Alexa Fluor 488 antibodies) for 1 h at room temperature. Slides were mounted with Vectashield and imaged using a Nikon Eclipse TE100. See Supplementary Data 4 for antibody information.

### Immunofluorescent microscopy

For detection of cytosolic ssDNA and dsDNA, cells growing on coverslips were washed with PBS 2 times, fixed with 4% paraformaldehyde in PBS for 30 min, and then permeabilized with 0.02% (for UM-SCC-1 and MDA1586 cells) or $2 \times 10^{-7}\%$ (for PCI-15B cells) saponin for 1 to 5 min and blocked with 5% bovine serum albumin (BSA) in PBS for 30 min. For ssDNA S1 nuclease (ThermoFisher Scientific, FEREN0321) and dsDNase (Life Technologies, #EN0771) treatment, cells were permeabilized with saponin for 2 min and treated with either nuclease for 10 min before fixation using 4% paraformaldehyde in PBS. The coverslips were then incubated with an anti-ssDNA or anti-dsDNA antibody overnight at 4 °C, washed, and then stained with a secondary antibody conjugated to an Alexa Fluor 647 or Alexa Fluor 477 anti-mouse IgG and DAPI (1 μg/mL) for 1 h at room temperature. Cells were then mounted on the slides with an antifade mounting medium (Vector Laboratories, #H-1000). Images were acquired by using an Andor Revolution XDi WD spinning disk confocal microscope. Imaris ×64 9.3.0 software was used to define the boundaries of cells and nuclei using the colors of GFP (MDA1586 and PCI-15B stable cell lines) or RFP (UM-SCC-1 stable cell line) and DAPI, then the mean fluorescence intensity of ssDNA/dsDNA in the cytosol of each cell was quantified. For detection of nuclear RPA32, p53, Rad51, and RelB, cells were fixed and permeabilized with 0.5% Triton 100 in PBS for 15 min, then blocked with 5% BSA-PBST (PBS + 0.1% Tween 20) for 30 min. Cells were incubated with each individual primary antibody overnight at 4 °C and then with a fluorescence-conjugated secondary antibody for 1 h at room temperature, mounted on the slides, imaged, and quantified as described above. Information on the antibodies used is provided in Supplementary Data 4.

### Quantification of cytosolic DNA

To isolate cytosolic DNA, $1 \times 10^7$ cells were lysed, and the nuclear, cytosolic, and mitochondrial fractions were obtained using a mitochondrial isolation kit for cell culture (ThermoFisher Scientific, #89874). In order to enable subsequent DNA purification, protease inhibitors were not used. Mitochondria were purified by centrifugation at $12,000 \times g$ to minimize their contamination of the cytosolic fraction. DNA was isolated from the nuclear, cytosolic, and mitochondrial fractions using a DNeasy blood and tissue kit (Qiagen, #69504), and dsDNA was quantified using an AccuBlue high-sensitivity dsDNA quantitation kit (Biotium, #31006).

### cGAMP measurement

Cells treated with or without HU were lysed by RIPA buffer with a 1× protease inhibitor cocktail. Cell lysates were used to measure cGAMP concentration using a 2′,3′-cyclic GAMP competitive ELISA kit (Invitrogen, #EIAGAMP) according to the manufacturer's protocol.

### In vitro invasion and migration assay

A 24-well BioCoat growth factor reduced Matrigel invasion chamber (Corning, #354483) was used for invasion and migration assays. Specifically, $5 \times 10^4$ UM-SCC-1 or $1 \times 10^5$ MDA1586 stable cells were seeded in the upper chambers and incubated with serum-free DMEM; DMEM supplemented with 10% fetal bovine serum was added to the lower chamber. The plates were placed in an incubator (37 °C) under 5% $CO_2$ for 24 h. The cells in the upper chamber were removed using cotton swabs, and the cells that had invaded and migrated to the lower chamber were stained with a solution of 0.2% crystal violet in 25% methanol. Then the chamber membranes were mounted on glass slides and examined with a Leica DMLA microscope. At least three random 100× fields were selected for each cell line. Cell numbers were counted and analyzed using Image J software (NIH).

### Animal study

Animal experiments were performed in accordance with protocols approved by the MD Anderson Cancer Center Institutional Animal Care and Use Committee (IACUC). For human cell line studies, stable cells were injected into the tail veins of 5- to 6-week-old athymic male nude mice (Harlan, athymic nu/nu) as previously described[81]. Mice injected with MDA1586 stable cell lines were fed a doxycycline-containing rodent diet (Envigo, #TD.01306). For murine 4T1 cell studies, stable cells were injected into the 4th mammary fat pads of 8-week-old female BALB/cJ or NOD SCID (NOD.Cg-Prkdcscid/J) mice (Jackson Laboratories). Tumor growth was measured by digital calipers, and tumor volumes were calculated using the formula $V = (L \times W \times W)/2$. At the end of the experiments, mice were sacrificed, primary tumor and/or lungs were collected and fixed, embedded in paraffin, sectioned, and stained with hematoxylin and eosin (H & E). To calculate the rate of microscopic metastasis (Fig. 4s, w and Supplementary Fig. 4o), 3 consecutive H & E-stained sections per lung, 100 to 300 μm apart, were selected, stained with H & E and scanned with an Aperio AT2 microscope (Leica) into an e-slide manager system. E-slides were viewed with Halo software (V2.3.2089.52; Indica Labs). The lung tissue section area, tumor nodule number, and tumor nodule size were measured by either manual or automatic methods. The manual method was used for tissues with countable metastatic nodules. Tumor nodules were identified and counted, and the area (mm²) of each nodule and the whole lung tissue (excluding large air ducts, large blood vessels, portal tissue, and lymph nodes, but including tumor tissue) was measured. Total tumor areas were summed, and the percentage of the lung area containing tumor was calculated. The automatic method was used for cases with tumor nodules that could not be counted and measured manually. In this case, the software's tissue annotation was set, and tissue classifiers for tumors, normal lungs, air

ducts, and blood clots were developed. Then the lung normal tissue area and tumor area were measured using these classifiers. Total tumor areas and tumor area percentages in the lung sections were calculated. To summarize the results, one of the 3 sections from each lung (total 8-10 lungs/group) was selected, and their lung metastatic rates were summed and averaged as one set. Shown in Fig. 4s, w and Supplementary Fig. 4o are the mean metastatic rates of 3 different sets (shown as 3 points) of H & E sections, in which each point represents the result from 8-10 individual sections from different lungs in each group.

### Immunohistochemistry and multiplex immunofluorescence staining

Paraformaldehyde-fixed, paraffin-embedded mouse lungs were sectioned. After deparaffinization and rehydration, antigens were retrieved by boiling in Dako target retrieval solution (#S1699) for 15 min. Endogenous peroxidase was blocked by incubating in 3% hydrogen peroxide for 30 min, and sections were then blocked by 5% BSA in Tris-buffered saline + polysorbate 20 (TBST) for 1 h at room temperature. For IHC staining, sections were incubated with primary antibodies diluted in 2.5% BSA in TBST at 4 °C overnight, then with a horseradish peroxidase-conjugated secondary antibody at room temperature for 1 h. Finally, results were visualized by using a Dako liquid DAB+ substrate chromogen system (K3468). For CD3[+], CD8[+], and RelB multiplex immunofluorescence staining, an Opal 7-color kit (Akoya Biosciences, #NEL811001KT) was used according to the manufacturer's protocol. See Supplementary Data 4 for antibody information.

### Multiplex immunofluorescent image acquisition and analysis

Immunofluorescent slides and adjacent H & E sections were scanned with the Vectra Polaris imaging system (Akoya Biosciences) following the manual's instructions, with high-power field scan (×40) using the fluorescent mode and bright field mode, respectively. The microscope captured the multispectral fluorescent spectra separately at the corresponding tyramide Opal fluorophore wavelength, with preset exposure times, and then these captures were stacked in 1 image (QPTiff) without disrupting the unique fluorescent spectral signature of the markers. The QPTiff image was analyzed in Visiopharm software for the regions of interest, necrosis, and tumor, for the entire section from selected tumors.

### Tuning strategies for cellular identification and phenotyping

All digitized images were analyzed using the Visiopharm software platform. Regions of interest (tumor and necrosis) were identified by a deep-learning algorithm using adjacent H & E sections and were subsequently overlayered to the corresponding fluorescent slides in the Visiopharm platform. For detecting nuclei, a pre-trained deep-learning algorithm available with the Visiopharm platform (U-Net architecture) was used. The convolutional neural network was trained to identify 3 components of the fluorescent images: (a) DAPI[+] nuclei; (b) boundaries of DAPI[+] nuclei; and (c) background. The algorithm magnification was set to 20× to maximize the ability to capture details in the images. Once nuclei in the sample were identified, the nuclear labels were expanded by 5 pixels in all directions to approximate the boundaries of cells, not just DAPI[+] nuclei, to define the cytoplasm. Finally, the cell segmentation was confirmed via visual inspection conducted by trained personnel.

For phenotyping cells, a targeted approach to generate the specific list of phenotypes (i.e., biomarker combinations) was used. Specifically, we were interested in finding phenotypes that were positive for a single biomarker (i.e., RelB, CD3[+], or CD8[+]) and double positive for 2 biomarkers (i.e., CD3[+]CD8[+]). For a given cell, the classification of each biomarker was gated using 2 independently controlled parameters: signal intensity and percent coverage. During the design of the

generalized classification algorithm, classification parameters were iteratively adjusted to maximize accuracy and minimize the occurrence of false positives and false negatives for each biomarker. Biomarker classifications were visually inspected and confirmed by multiple researchers. Once the parameters for accurate classification were optimized, those settings were applied to all images. Once the algorithms were applied to the images, a list of output variables, including counts of each identified phenotype per region (i.e., tumor and necrosis) was generated. For quantification of the RelB nuclei/cytoplasm ratio, 10 to 12 regions in the center of the tumor region from 3 sections from 3 different tumors per group (in total 57,370 and 78,073 cells) were randomly selected (Fig. 5d), and the spatial location in Cartesian coordinates (e.g., center x and center y coordinates) for each cell on the whole slide were generated. RelB mean fluorescent intensity (MFI) of the nuclei and RelB MFI of the whole cell were extracted using Visiopharm for all the cells found in the region of interest, and the RelB nuclei/cytoplasm ratio was calculated using the following formula:

$$\text{Relative RelB Nucei/Cytoplasm Ratio} = \left( \frac{\text{Nuclei RelB MFI} - \text{Whole Cell RelB MFI}}{\text{Whole Cell RelB Mean Intensity}} \right) * 100 \quad (1)$$

For T-cell phenotyping, the entire tumor area from each tumor section (approximately $0.3$–$1.7 \times 10^6$ cells/section, a total of 4 tumor sections from 4 different tumors per group) was selected (Fig. 5f, g).

### Multiplexed ion beam imaging technology

Multiplexed ion beam imaging (MIBI) technology and service provided by IONPATH (Menlo Park, California) were used for NK and dendritic cell staining (Fig. 5h–m). For full details of the MIBI methods, see the companion paper[84]. Briefly, antibodies were conjugated to isotopic metal reporters. Formalin-fixed paraffin-embedded tissue sections were stained with metal isotope-labeled antibodies and then imaged using time-of-flight secondary ion mass spectrometry. The masses of detected species are then assigned to target biomolecules given the unique metal isotope label of each antibody, creating multiplexed images. From each group, 12–13 regions of interest (ROIs) (800 μm × 800 μm/ROI) of tumor areas were selected from each group (3 tumors/group, 4–5 ROIs/tumor) using the corresponding H&E images for guidance. Multiplexed image sets were extracted, slide background-subtracted, denoised, and aggregate filtered. Cell segmentation takes advantage of the multiplexed nature of MIBI data by combining the nuclear dsDNA signal with cytoplasmic and membrane markers to accurately delineate and identify single cells in the tissue image dataset. Cell classification was performed with a machine learning framework that exploits the morphology and intensity of biomarkers to identify positive regions. For the markers present in the panel, a set of deep-learning models was trained based on expert annotation on a subset of data. These annotations capture the variability in staining patterns of the biomarkers in the tissues. Once trained, the models were applied to all the images to yield a model score for every segmented cell to be positive for biomarkers used to phenotype cells. A probability threshold was chosen by the pathologist for each biomarker based on a manual review of images. Cells with a probability greater than the threshold were called positive for that biomarker. Expression of markers was quantified at the single-cell level using summed intensities. The accuracy of cell classification was visually verified by a pathologist. See Supplementary Data 4 for MIBI antibody information.

### Bulk RNA sequencing and analyses

Triplicated bulk RNA was isolated from each sample using the RNeasy Plus Mini Kit (Qiagen, # 74134), and then sequenced by Novogene using an Illumina HiSeq4000 sequencer. The quality of the raw FASTQ

files was checked with the FASTQC package (https://www.bioinformatics.babraham.ac.uk/projects/fastqc/). The TopHat package (2.1.1)[85] or STAR (2.7.9a) was used to align the RNAseq data to the human reference genome GRCh38 to generate bam files. The bam files were sorted by Samtools software (1.8)[86]. The sorted bam files were then converted to SAM files. HTSeq software (0.11.0)[87] was used to count the number of reads falling in the exonic regions of each gene. The read counts represented the RNA expression level of each gene or region. We combined the HTSeq read count results for each sample to obtain the raw gene expression matrix data. We examined the matrix to filter out genes with low overall read counts across samples (≤ 5 reads in all of the samples), which were not included in our group comparisons. The data were normalized using the Variance stabilizing transformation in the DESeq (1.34.0) package for R[85]. To identify differentially expressed genes, a 2-sample $t$-test was used to determine whether there was a significant difference between each gene's mean expression level in the 2 groups. To adjust for multiple tests, we applied the Benjamini–Hochberg procedure[86]. Genes with a false-discovery rate (FDR) ($q$-value) less than or equal to 0.1 were considered to be significantly differentially expressed.

GSEA was conducted together with analysis of the molecular signatures database (MSigDB) to determine biologically meaningful gene sets that were enriched between the 2 biological conditions in the comparisons[88] (https://www.gsea-msigdb.org/gsea/index.jsp). GSEA analysis was conducted by using the Hallmark gene set (h.all.v7.0.symbols.gmt) as the gene set database[89] (https://www.gsea-msigdb.org/gsea/msigdb/human/genesets.jsp?collection=H) and performing 1000 permutations. GSEA calculates an enrichment score (ES) for each gene set from the Hallmark database by ranking the user gene list based on its degree of overrepresentation in one of the 2 groups being compared. Highly ranked genes are overrepresented in one group, while lower-ranked genes are overrepresented in the second one. The degree of overrepresentation of each Hallmark gene set among higher or lower-ranked genes defines its ES. A positive ES indicates enrichment of a given gene set in one group, and a negative ES indicates enrichment in the other group. To account for differences in the size of the gene sets and in other variables and allow comparisons among gene sets, the ES was normalized to a normalized ES (NES). In this study, we used the NES to analyze and represent our data. Gene sets were considered significantly enriched at an FDR $q$-value < 0.25.

### The Cancer Genome Atlas (TCGA) analyses

Data from OSCC patients from TCGA were used to evaluate our molecular findings in a clinical context. Clinical data were retrieved from Liu et al.[90]. The TCGA MC3 head and neck cancer mutation data (MAF file)[91] was retrieved using the R packages PoisonAlien/TCGAmutations[92] (https://github.com/PoisonAlien/TCGAmutations) and MAFtools[93] (https://github.com/PoisonAlien/maftools). RNAseq data were recovered from the National Cancer Institute's Genomic Data Commons (GDC) (https://portal.gdc.cancer.gov/)[94]. We included in our in silico cohort 221 patients with oral cavity tumors that were HPV-negative and *TP53* mutant and had whole-exome and RNAseq data available. To validate our in silico findings, we retrieved genomic and clinicopathological data from squamous carcinomas from TCGA[95,96] (https://gdc.cancer.gov/about-data/publications/PanCan-Squamous-2018). We selected lung ($n = 399$) and larynx ($n = 100$) squamous cell carcinoma samples, exhibiting *TP53* mutations, and negative for HPV infection[95].

EMT scores for the selected samples were computed based on a pan-cancer EMT signature. The signature was derived from TCGA RNAseq data consisting of 11 tumor types ($n = 1934$ tumors). The score was calculated by taking the difference between the average of mesenchymal gene expression and the average of epithelial gene expression[97]. Immune features for the selected samples were retrieved from Thorsson et al.[98], which reported immunogenomic characterization of all TCGA tumors. Immune enrichment scores were calculated and considered as continuous variables in our analyses. In order to estimate the enrichment of specific Hallmark gene sets (h.all.v7.0.symbols.gmt) (https://data.broadinstitute.org/gsea-msigdb/msigdb/release/7.0/) among the TCGA samples, we used ssGSEA, an extension of GSEA. The algorithm calculates separate ESs for each pairing of a sample and gene set. Each ssGSEA ES represents the degree to which the genes in a particular gene set are coordinately upregulated or downregulated within a sample. The calculated ssGSEA scores for each Hallmark gene set were treated as continuous variables in our analysis.

The generated data were analyzed using JMP 15.0 software. Hierarchical clustering analysis was conducted using the Ward method for defining distances between clusters. Associations between categorical variables and 5-year overall, disease-specific, and progression-free survival were evaluated with the log-rank test. Associations between categorical and continuous variables were tested by the Kruskal–Wallis test followed by post hoc analysis using the Steel-Dwass test.

### Statistics and reproducibility

Statistical methods are provided with the figure legends or figures. All the western blots shown in the paper are representative images from at least three independent experiments.

### Reporting summary

Further information on research design is available in the Nature Portfolio Reporting Summary linked to this article.

### Data availability

Source data for Figs. 1–6 and Supplementary Figs. 1–8 are provided with this paper; TCGA genomic and clinical data is available through the National Cancer Institute's Genomic Data Commons web portal (https://portal.gdc.cancer.gov/); SILAC MS/MS proteomic mass spectrometry proteomics data generated in this study have been deposited in the ProteomeXchange Consortium via the PRIDE partner repository with the accession number PXD047094; bulk RNAseq data generated in this study have been deposited in NCBI's Gene Expression Omnibus under GEO series accession number GSE164433. Source data are provided in this paper.

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

## Acknowledgements

The authors would like to thank Amy Ninetto, Scientific Editor, Research Medical Library, MD Anderson Cancer Center, for editing the manuscript; This research was partly performed in the Flow Cytometry & Cellular Imaging Core Facility, which is supported in part by the National Institutes of Health through M. D. Anderson's Cancer Center Support Grant P30 CA016672. This work was also supported by National Institutes of Health grants (1 R50 CA243707-01A1, J.K.B., R01-DE014613, J.W., R01-DE028061, C.R.P., R01-CA216437, J.C., R01-DE024601, R01-DE014613 and R01-DE030875, J.N.M., R01-DE014613 and R01-DE030875, G.Z.), and by MD Anderson Cancer Center Institutional Research Grant (600382-30-122489-21, G.Z.). The funders had no role in study design, data collection and analysis, decision to publish, or preparation of the manuscript.

## Author contributions

G.Z. and J.N.M. conceived the project. M.Z. and T.W. performed most of the experiments. W.J. and M.A.G. provided research assistance. F.O.G., W.M. and L.S. analyzed TCGA and RNAseq data; F.O.G. performed statistical analyses; Z.C. performed SILAC assays; D.J.M. performed DNA fiber assay. J.A.G. performed multiplex IHC data analyses; Q.W. performed MIBI data analyses; X.T. scored the mouse lung metastases; A.S.M. performed chromosome spread assay and analyses; S.P., M.G.R., J.K.B., S-Y.L., J.W., C.R.P., J.C., J.N.M. and G.Z. oversaw and interpreted data. G.Z. wrote the manuscript with input from all authors. All authors read and approved the final manuscript.

## Competing interests

The authors declare no competing interests.

## Additional information

[1]Department of Head and Neck Surgery, The University of Texas MD Anderson Cancer Center, Houston, TX 77030, USA. [2]Department of Head and Neck Surgery, Key Laboratory of Carcinogenesis and Translational Research, Peking University Cancer Hospital & Institute, 100142 Beijing, China. [3]Department of Experimental Radiation Oncology, The University of Texas MD Anderson Cancer Center, Houston, TX 77030, USA. [4]Department of Systems Biology, The University of Texas MD Anderson Cancer Center, Houston, TX 77030, USA. [5]Department of Leukemia, The University of Texas MD Anderson Cancer Center, Houston, TX 77030, USA. [6]Department of Bioinformatics and Computational Biology, The University of Texas MD Anderson Cancer Center, Houston, TX 77030, USA. [7]Department of Translational Molecular Pathology, The University of Texas MD Anderson Cancer Center, Houston, TX 77030, USA. [8]Department of Genetics, The University of Texas MD Anderson Cancer Center, Houston, TX 77030, USA. [9]Present address: Center for Immunotherapy and Precision Immuno-Oncology, Cleveland Clinic, Cleveland, OH 44195, USA. [10]Present address: Lerner Research Institute, Cleveland Clinic, Cleveland, OH 44195, USA. [11]Present address: Department of Surgery—Otolaryngology, Yale School of Medicine, New Haven, CT 06250, USA. [12]These authors contributed equally: Mei Zhao, Tianxiao Wang. ✉e-mail: jmyers@mdanderson.org; gzhou@mdanderson.org

