## [Peer Review File · Nature Communications]

Mutant p53 gains oncogenic functions through a chromosomal instability-induced cytosolic DNA responseREVIEWER COMMENTS

Reviewer #1 (Remarks to the Author):

In this manuscript, Zhao et al investigated the oncogenic gain of function (GOF) role of mutant p53. They identified interaction between GOF mutant p53 with DNA licensing factor MCM5 in which it was shown that the inability of mutant p53 to bind to MCM5 resulted in elevated replication stress and consequently chromosomal instability (CIN). Zhao and colleagues demonstrated that one of the cellular consequences for the increased of CIN level is the accumulation of cytosolic DNA in which the later leads to activation of cGAS-STING pathway. In line with previously published findings, the authors showed the activity of cGAS-STING stimulates the non-canonical NF- κ B signaling, promoting cellular invasion and metastasis in vitro and in vivo. Furthermore, it was shown that the activation of non-canonical NF- κ B leads to the silencing of interferon signaling and an immunosuppressive tumor microenvironment observed by decreased in CD3+ CD8+ T-cells.

This work is important, widely relevant and the experiments described are rigorously conducted. Given the prevalence of mutant p53 in human cancer, the work by Zhao and colleagues is an important and relevant one that mechanistically dissects the link between gain of function mutant p53, tumor microenvironment, and metastasis. The authors elegantly showed that accumulation of cytosolic DNA because of chromosomal instability suppresses interferon signaling and endorses the dissemination of metastasis via cGAS-STING-NC-NF- κ B signaling. While we think this study is interesting and well supported with the current published works on chromosomal instability and metastasis, we have listed below some areas of constructive criticisms:

1) The authors proposed a mechanism where mutp53 induced CIN through interacting with MCM5 with evidence including that the mutp53 phenotype can be rescued by MCM5 overexpression. It is possible that mutp53 induces CIN through other mechanisms, and MCM5, when it is overexpressed, artificially titrates/blocks mutp53, instead of working downstream of MCM5. The author should test whether overexpressing non-functional, but mutp53 interacting MCM5 can still suppress mutp53 phenotype.

2) The authors showed accumulation of cytosolic DNA in mutp53 expressing cells leading to activation of cGAS. Some direct measurement of cGAMP in cell lysates should be performed to show the increase of cGAS activity in mutp53 cells.

3) The authors claimed that interferon signaling was not activated in response to CIN-induced mutp53. While they showed that there was increased localization of RelB in tumors with mutp53, it would be helpful to further explore the behavior and response of p65 and IRF3 to STING activation in these cells. The assumption based on their data is that the signaling is primarily RelB however it would be helpful to further support this hypothesis with additional cell-based experimental data.

Reviewer #2 (Remarks to the Author):

In this elegant study, the authors investigate the contribution of different gain of function (GOF) mutations in p53 to genomic instability. Using mass spectrometry, they find that several, but not all forms of mutant bind to MCM5. They also describe that cells with mutp53 have increased numbers of chromosomal abnormalities, more RPA32 foci and more cytosolic ssDNA and dsDNA, which is partly alleviated by ectopic overexpression of MCM5. Furthermore, they find that the genomic instability caused by GOF-mutp53-MCM5 interaction further triggers NC-NF κ B inflammatory signalling, which promotes metastasis by modulating the tumor microenvironment.

Overall, this is a well-designed and well-executed study. However, there are a few concerns that the

authors might want to address before publishing this study.

- The authors show that several mutant versions of p53 bind to MCM5, except for R248Q. They do not comment on why some mutants do bind and others do not. While it might be outside of the scope of this study to determine why exactly which mutants bind to MCM5 and which ones do not, the authors should comment on this.
- Related to this: why as a DeltaDBD (I assume a non-DNA binding version of p53) bind much more MCM5. This control (?) is not described in the manuscript (Fig 2C, lane 11), unless I overlooked it.
- The authors present various Western blots in which RPA32 shows altered chromatin binding in p53 mutant cell lines and that this changes following overexpression of MCM5 or knockdown of MCM5 (in cell lines in which mutant p53 is knocked down). While the blots illustrate the findings well, the authors might want to quantify the effect and include multiple biological replicates to ensure this effect is consistent and to determine the average effect size.
- Related to this, in e.g. Fig 2b, the authors observe an extra RPA32 band, but only in the chromatin-bound fraction. What is this second band: a phosphorylation product?
- Similarly, the authors show multiple IF experiments in which RPA32 is localized in granules (vs. homogenous nuclear localization). This effect should also be quantified for multiple cells and in biological replicates.
- The authors use HU to induce replication stress and determine cytoplasmic ssDNA and dsDNA. While they nicely show and quantify this using IF and DNA-recognizing antibodies, the authors do not quantify micronuclei, which is another important instigator of cGAS-STING signaling. As micronuclei are a common outcome of replication stress and their rupture will trigger cGAS, the authors should also quantify these in the various genotypes and following HU treatment.
- The authors quantify CIN using metaphase spreads and show clear effects. Adding time-lapse data of cells going through mitosis would be a complementary technique to strengthen this point and help determine whether the abnormalities in the metaphase spreads (partly) result from mitotic abnormalities that are secondary to the replication stress.
- Authors discuss the role of cGAS-STING pathway in the induction of NF- κ B signaling and all the downstream effects. However, all the experiments relating to this pathway are done in the context of shSTING or shRelB. It has been previously shown that STING can be activated independently of cGAS. Therefore, authors might want to show the role of cGAS in their phenotype as well.
- Where applicable, authors should more clearly show the individual replicate values for each experiment (i.e. Fig 4f, 4h, m, q, 5d,...)
- In figure 4b, there is a clear increase in p53 expression upon knockdown of RelB. Why is this? The authors should comment on that.
- In figure 4c, a p53 blot is missing.
- In Figure 4, authors show invasion assays using R273H mut UM-SCC1 cells in combination with shRelB or shSTING. Authors also show similar results G245D mut UM-SSC-1 using shRelB. Authors should also show the effect of shSTING in invasion capacity of G245D mut UM-SSC-1 cells.
- In Fig4k, there appears to be some mislabeling of the genotypes in the schematic representation. For instance, in "(5) R270H" should be replaced for "(5) R273H" and in "(6) R273H-shSting1" should be replaced for "(6) R273H-shRelB2"

- In Fig5f and 5g, authors should perform and include CD3 and CD8 quantification from at least 8 (out of the 11) primary 4T1 tumors. This will make their data more robust.
- In the transcriptome studies (Fig 6a, b), we miss RelB KO cells in a wildtype background (i.e. without p53 GOF mutations).
- Using TCGA data, authors showed that tumors with low scores on IFN signaling but high inflammatory signatures have a lower NK cell activated signature and lowered follicular helper signature. Is this trend also observed in their vivo experiments?
- Could authors generalize their findings of 4 distinct groups of patients with IFN high/low or inflammation high/low OSCC to other cancer types with mutp53 using TCGA?
- The authors show that NC-NFKB inhibits IFN signaling. Understanding how this works mechanistically would add significantly to the impact of the study. If this is beyond the scope of this study, the authors should at least discuss how they think this works.

Reviewer #3 (Remarks to the Author):

In the paper, the authors try to find the mechanism by which gain-of-function of mutant p53 proteins promotes tumor development and progression. After finding p53 interactors, the authors focus on MCM5 and show how the mutant p53-MCM5 interaction drives chromosomal instability which leads to STING-dependent response and activation of non-canonical NFkB signaling. These in turn promote tumor cell metastasis and an immunosuppressive tumor microenvironment. The authors investigate the chain of reactions leading to mutant p53 GOF phenotype in an orderly fashion, yet some information is missing and should be addressed by the authors as described below.

Immunoprecipitation and mass spectrometry analysis:

1. The authors use for the IP two different p53 antibodies: DO-1 and Pab240. According to the manufacturer's website, Pab240 does not bind WT p53 in its native form, only the mutant protein (<https://www.scbt.com/p/p53-antibody-pab-240>, "p53 Antibody (Pab 240) is recommended for detection of only mutant p53 under non-denaturing conditions but is equally reactive with mutant and wildtype p53 under denaturing conditions"). Is this difference in reactivity was taken into consideration in the experimental design? The authors should state these differences in the text and methods section.
2. Supplementary table 1 – please provide titles to the different experiments, it is not clear which sample is heavily labeled or not, and which cells were used or compared in each experiment.
3. According to the methods section, the authors used for some of the IP experiments cells with PIK3CA H1047R mutation. It is only mentioned in the methods section and not explained why this mutation was included and how it affects the IP results.
4. From the IP experiment design, it is not possible to know which interactors are preferentially binding to the mutant or WT p53 protein. Overexpressing each form separately and performing the IP can solve it. In the IP experiment, both WT and mutant forms are probably immunoprecipitated (as it is not indicated that the endogenous WT p53 was knocked out). Alternatively, if one of the antibodies binds only to the mutant form maybe the authors can use it for showing the differential binding of interactors to the mutant compared to the WT p53.
5. What is the ratio of WT and mutant p53 protein expression in the cells used in the study? Making a table that includes the mutation/WT p53 status of each cell that was used in the study, as well as the relative protein abundance of the mutant/WT p53 protein and MCM5 protein, can help the readers understand the effect of their abundance on the observed phenotypes and make an easy to follow guide for the different cells.
6. The authors should provide validation of protein overexpression in the cells, by western blot or any

other means.

7. The methods section is lacking a lot of information including how IP samples were prepared for mass spectrometry analysis, there is no information at all about the mass spectrometry analysis (which instrument? Which setting? And so on), what were the setting for the MaxQuant search? Which MaxQuant version was used? Did the mutation sequence was included in the database? How the data was processed and normalized?

8. Data deposition: The authors provided the user name but not the password for reviewing their data deposition in PRIDE. Please make sure to include in addition to the search and raw files also the fasta file that was used for the search, the MaxQuant installation file of the version that was used for analyzing the data, and a file that describes the raw file name with the sample name. Please provide in the revised version of the manuscript the password for the data deposition so it could be accessed and reviewed.

IP results:

1. The authors identified 33 different proteins that interact with mutp53, but except for listing them, no further information is provided. Do the authors see any pathway/function enrichment for the interactors? Do other interactors were derived from protein complexes (except MCM proteins)? Which ones are already known and which are new interactions?

2. The authors should describe in the text why they decided to focus only on MCM5 out of all the interactors that were identified? What was the rationale behind their choice?

3. Did the authors validate all unknown interactions as well (using WB or any other approach)?

Validation of mutp53-MCM5 interaction:

1. For all WB images, the authors should provide also uncropped blots.

2. Why the validation of the interactions with MCM5 and other MCM (as in figure 1) were not done in the cells in which the IP experiments were performed (UM-SCC-1)?

3. Figure 1 and Extended Figure 1:

a. Please indicate in the figure legend that p53 was KD in the cells as control (it is only indicated in the figure as shp53).

b. Why in some of the cells p53 was KD while other cells treated with IFN γ ?

4. Page 6: "although R273H/L and several other mutp53s also interacted with other components of the MCM2-7 complex (e.g., MCM2, MCM3, and MCM7), they bound most strongly to MCM5" – did the data normalized to each MCM expression level, can it be that the enrichment in pull down result from different MCM protein expression and not binding affinity?

Other comments:

1. Page 4: "This finding strongly suggests that many previously reported mutp53 GOF phenotypes are actually consequences of CIN in mutp53 cells" – this statement should be accompanied by analyzing previous data to show it, or the text should be adjusted to suggest that this is the author's assumption.

2. Page 5: "Here, we examined the mutp53 interactome and identified MCM5 as a direct covalently bound protein target through which mutp53 exerts its GOF activities to predispose cancer cells with mutp53 to replication stress and CIN" – the authors do not demonstrate in the study that MCM5 and mutp53 are bound covalently or that MCM5 and mutp53 bind directly. The IP experiments in the manuscript do not prove any of these claims, please adjust the text or perform the required experiments to prove it.

3. Why TCGA analysis was done only for HPV-negative OSCC patients? do the observations in the study are relevant only to this cancer type and not for any mutant p53 cancer types?

Reviewer #4 (Remarks to the Author):

In the article 'Mutant p53 gains oncogenic functions through a chromosomal instability-induced cytosolic DNA response' the authors look at how gain-of-function mutants are linked to CIN in various cancer cells. They identified a role for mutant p53 in binding MCM5 and promoting a cytosolic DNA accumulation that promotes the STING pathway and suppresses the IFN release. The manuscript is

well-written and the figures are clear. Mutant p53 has been associated to Chromosomal instability and in this manuscript the authors are addressing a novel pathway through which mutant p53 acts on this. This work is therefore a nice novel addition to previous work by others.

Although each individual figure is clear and draws the right conclusions, the work using the patient data does not really support the first part of the paper in which mutant p53 is regulating replication stress through MCM5 binding. These data merely address whether an interferon response is correlated the NF- κ B activity and survival, but do not take into account mutant p53 status or MCM5 activity. Statistics and the use of error bars (SD or SEM) in the paper are a bit random and leave some questions.

Major comments:

1. None of the data in patients are actually linked to p53 status or MCM5 expression and are addressing a different question than rest of the manuscript. These data address whether interferon signalling is inversely correlated to survival or NF- κ B activity. The authors select for patients with mutant p53 status. All mutations or only point mutants? What mutants are selected? A 248Q mutations is relatively frequent in OSCC and should not activate this pathway as that one does not bind MCM5. If selecting only 248Q patients is this actually the case? Are there enough patients to do this in the OSCC groups?
2. Does a loss of Sting in 248Q cells still result in a RelB depletion in the nucleus (fig 4A)? To prove that the Sting pathway is downstream the mutant p53/MCM5 interaction
3. RPA32 is used as a marker of replication stress. The authors look in the chromatin fraction and have indicated both bands as markers for RPA32 binding to chromatin. I presume the higher band is p-RPA32, which many in the field use as marker for replication stress. Clearly this phospho-band is following the pattern as described in the results section for figure 2A and 2B. The bottom RPA-32 band does not and stays relatively equal in fig 2B. In Figure 2C no top band is seen and the authors make the conclusions about the bottom band. Can the authors clarify they are looking at both bands and reference that binding to chromatin regardless of phosphorylation of RPA-32 is a marker of replication stress.
4. The authors use a lot of different cell lines and it is not always clear why certain cell lines are used for different purposes. Some clarification is needed in the text
5. In figure 4m the authors show a bar graph with very small error bars. The supplemental data show that some mice did not develop any metastases. It therefore seems very unlikely such small error bars could be achieved if taken into account the numbers of mice not developing any metastases. More than half of the mice in the R270H shRelB 1 didn't develop mets. The graph shows 10% with an error bar of less than 1%. Surely half of these datapoints would have been 0 (no mets). To then get to 10% the remaining 5 mice must have had an area of 20% or more with mets to get to a 10% mean (suggested in legend) which would result in very large error bars. Statistics is unclear
6. Error bars in most figures are very small despite large variations seen in raw data and between samples. Mainly concerning in the in vivo data. Are the authors sure these depict SD and not SEM? Very confusing that different error bars are used in different figures and even parts of figures. Why different statistics test in 6g and h compared to I and J in the same figure?
7. MCM5 itself is causing chromosomal instability and ssDNA and dsDNA accumulation in the cytoplasm. However, in the presence of mutant p53, it inhibits the mutant p53 mediated accumulation. It is not explained how this works.

Dear Reviewers:

We greatly appreciate all the time that you have spent to review our work and for your constructive comments and suggestions to make this manuscript scientifically stronger. Since receiving your critiques, we have followed your recommendations and systematically designed additional experiments shown in this revised manuscript to address the concerns that you raised.

Compared to our initial submission, a total of 33 new additional figure panels (Fig. 1a; Extended Data Fig. 1f-h, 1m-o; Extended Data Fig 3o,q; Fig. 4a,c,e, g,h,k,l; Extended Data Fig. 4g,h; Fig. 5h-m; Extended Data Fig. 8a-j) have been included in this revised manuscript. The Data Sources of each figure panel including the full scan of all the western blots shown in the manuscript have also been provided. In addition, all the changes and modifications in the revised manuscript have been tracked and highlighted to display any differences with the original manuscript that we previously submitted.

While we were revising this work, an exciting paper from Dr. Bakhom and colleagues was published (Li, J. *et al.* **Non-cell-autonomous cancer progression from chromosomal instability**. *Nature* **620**, 1080-1088 (2023)), which confirmed our findings that demonstrated the importance of chromosomal instability-cGAS-Sting signaling in immunosuppression and metastasis, our work further elucidated that how p53 mutations are implicated in this process. Therefore, both Dr. Bakhom & colleagues' work and our work highlight the significant importance of this topic in understanding the molecular mechanisms of tumor development and progression. In addition, Reviewers #2 and 4 raised a critical question about how to reconcile role of hot-spot R248Q mutation that did not bind to MCM5. Here, we provide additional evidence from our mass spectrometry results demonstrating that R248Q binds to other MCM subunits such as MCM7. Through the similar mechanism as we proposed, the R248Q mutation also achieves gain-of-function activity (see the following response in detail). Therefore, we believe our work, together with the recent work from Dr. Dr. Bakhom's group, have revealed one of the most important insights into p53's genome guardian role in tumor development and progression.

Based on all the information of our much enhanced and significantly revised manuscript, we hope that our manuscript will be accepted to be published in *Nature Communications*. We thank you very much for your comprehensive review and constructive feedback with suggestions that have enhanced our work and the significance of our findings.

Here are our point-by-point responses to the reviewer comments:

REVIEWER COMMENTS

Reviewer #1 (Remarks to the Author):

In this manuscript, Zhao et al investigated the oncogenic gain of function (GOF) role of mutant p53. They identified interaction between GOF mutant p53 with DNA licensing factor MCM5 in which it was shown that the inability of mutant p53 to bind to MCM5 resulted in elevated replication stress and consequently chromosomal instability (CIN). Zhao and colleagues demonstrated that one of the cellular consequences for the increased of CIN level is the accumulation of cytosolic DNA in which the later leads to activation of cGAS-STING pathway. In line with previously published findings, the authors showed the activity of cGAS-STING stimulates the non-canonical NF-kB signaling, promoting cellular invasion and metastasis in vitro and in vivo. Furthermore, it was shown that the activation of non-canonical NF-kB leads to the silencing of interferon signaling and an immunosuppressive tumor microenvironment observed by decreased in CD3+ CD8+ T-cells.

This work is important, widely relevant and the experiments described are rigorously conducted. Given the prevalence of mutant p53 in human cancer, the work by Zhao and colleagues is an important and

relevant one that mechanistically dissects the link between gain of function mutant p53, tumor microenvironment, and metastasis. The authors elegantly showed that accumulation of cytosolic DNA because of chromosomal instability suppresses interferon signaling and endorses the dissemination of metastasis via cGAS-STING-NC-NF-κB signaling. While we think this study is interesting and well supported with the current published works on chromosomal instability and metastasis, we have listed below some areas of constructive criticisms:

Our response: We are deeply appreciative of this reviewer's recognition of the importance of this work. Thank you very much!

1) The authors proposed a mechanism where mutp53 induced CIN through interacting with MCM5 with evidence including that the mutp53 phenotype can be rescued by MCM5 overexpression. It is possible that mutp53 induces CIN through other mechanisms, and MCM5, when it is overexpressed, artificially titrates/blocks mutp53, instead of working downstream of MCM5. The author should test whether overexpressing non-functional, but mutp53 interacting MCM5 can still suppress mutp53 phenotype.

Our response: The reviewer raised an excellent point, and we developed the recommended non-functional, but mutp53 interacting MCM5 and found that it can still suppress mutp53-associated phenotypes.

We initially also recognized the importance of this issue and provided MCM2 that had a weak R273H mutp53 binding as a control to demonstrate the specificity of MCM5-mutp53 interaction and its consequent impacts (Extended Data Fig1l, previously Extended Data Fig1i). With the reviewer's suggestion to test "whether overexpressing a non-functional, but mutp53 interacting MCM5 can still suppress mutp53 phenotype", we generated such mutant MCM5 construct (MCM5 R732A & K734A) (Extended Fig. 1m, n), and showed that this mutant MCM5 inhibited mutp53-mediated phenotypes even better than wild-type MCM5 (Extended Data Fig1o; Extended Data Fig3o,q and Extended Data Fig. 4g,h). The detail explanation and reference of these results are described in the text lines 177-188, 227-231, 255-259 and 276-279.

2) The authors showed accumulation of cytosolic DNA in mutp53 expressing cells leading to activation of cGAS. Some direct measurement of cGAMP in cell lysates should be performed to show the increase of cGAS activity in mutp53 cells.

Our response: Thank you very much for the suggestion and this result was provided in Extended Data Fig. 3o and described in text lines 225-231.

3) The authors claimed that interferon signaling was not activated in response to CIN-induced mutp53. While they showed that there was increased localization of RelB in tumors with mutp53, it would be helpful to further explore the behavior and response of p65 and IRF3 to STING activation in these cells. The assumption based on their data is that the signaling is primarily RelB however it would be helpful to further support this hypothesis with additional cell-based experimental data.

Our response: yes, we also examined the impact of mutp53 on the response of p65 and IRF3 to replication stress in different cells (Extended Data Fig. 3p,r,s). However, p65 and IRF3 seemed to be only affected in MDA1986 and PCI-15B cells when their endogenous mutp53s were knocked down (Extended Data Fig. 3r,s)(assume STING is intact in these cells), but not affected by overexpression of mutp53 in p53-null UM-SCC-1 cells in which only NC-NF-κB was constitutively activated by expression of mutp53 regardless of HU treatment (Extended Data Fig. 3p). One of the possibilities of no responses of p65 and IRF3 in UM-SCC-1 cells could be the lack of intact STING signaling in these cell lines. However, this possibility is not likely since our results clearly showed that NC-NF-κB activation in UM-SCC-1 cells are dependent on STING (Fig. 4b,c), indicating that STING signaling is intact (active) in these cell lines.

Therefore, as we described (in text lines 251-255), although p65 and IRF3 behave differently in different cells, activation of NC-NF- κ B was the most common event that occurred in all the cell lines that we tested. In addition, given the differences in duration between canonical NF- κ B (acute but transient) and NC-NF- κ B (slow but persistent) as we discussed (text lines 425-429), we believed that NC-NF- κ B is one of the main signaling events when cells are subjected to chronic CIN, which leads to low level but sustained cGAS-Sting activation.

Once again, thank you for your positive comments and thoughtful comments about our work. We feel that this helpful input has enabled us to enhance the manuscript.

Reviewer #2 (Remarks to the Author):

In this elegant study, the authors investigate the contribution of different gain of function (GOF) mutations in p53 to genomic instability. Using mass spectrometry, they find that several, but not all forms of mutant bind to MCM5. They also describe that cells with mutp53 have increased numbers of chromosomal abnormalities, more RPA32 foci and more cytosolic ssDNA and dsDNA, which is partly alleviated by ectopic overexpression of MCM5. Furthermore, they find that the genomic instability caused by GOF-mutp53-MCM5 interaction further triggers NC-NFKB inflammatory signalling, which promotes metastasis by modulating the tumor microenvironment.

Overall, this is a well-designed and well-executed study. However, there are a few concerns that the authors might want to address before publishing this study.

Our response: We deeply appreciate reviewer's comments and suggestions to enhance our study.

• The authors show that several mutant versions of p53 bind to MCM5, except for R248Q. They do not comment on why some mutants do bind and others do not. While it might be outside of the scope of this study to determine why exactly which mutants bind to MCM5 and which ones do not, the authors should comment on this.

Our response: This is an excellent question, that drove us to further explore the interaction of different mutant p53 forms with other MCM family members. As we briefly discussed above, in our original mass spectrum data, in addition to MCM5 we also pulled out MCM7 (supplementary table 1), despite that our current study mainly focused on the MCM5's role given that it has a strong interaction with R273H mutp53. Herein we provided additional evidence showing that although R248Q mutp53 did not bind to MCM5 well (Fig. 1d, lane 7 and Extended Data Fig. 1f, lane 6), it strongly bound to MCM7 (Extended Data Fig. 1g, lane 4). In addition, R248Q mutp53 exhibited a greater potential to induce replication stress (Extended Data Fig. 1h, lanes 6 vs 4) and *in vitro* transwell invasion (data not shown) than did R273H mutp53 in UM-SCC-1 cells despite that its expression level was lower than that of R273H mutp53 (Extended Data Fig. 1h, lanes 5 & 6 vs 3 & 4). Therefore, through the same mechanisms (i.e., mutp53-MCMs-CIN-cGAS-Sting-NC-NF- κ B signaling) by interacting with and targeting different members of MCM2-7 subunits such as MCM7, R248Q mutp53 is expected to predispose cells to CIN and achieve the GOF activities as we described in this study. Our current study using R273H-MCM5 was an example of a "proof-of-principle" study that demonstrated a novel mechanism involved in mutp53 gain-of-functions. We have included all these new information in text lines 145-149, 159-164, 167-171, 438-452. Similarly, in addition to its interaction with MCM5, R273H was also shown to interact with other members of MCM2-7 subunits (Fig. 1f,g), that likely can impact MCM2-7 functions, as well. Therefore, based on all our results, we have generalized our hypothesis to MCMs and not only restrict our hypothesis to MCM5 (see Abstract and Fig. 6k),

We agree with that it is unrealistic for us to determine which mutant binds to which MCMs in current study given that p53 mutants are many proteins with different properties. However, our finding of mutp53-MCMs interaction is consistent with and supported by a recent study (Schaefer-Ramadan, S., Aleksic, J., Al-Thani, N. M. & Malek, J. A. **Novel protein contact points among TP53 and minichromosome maintenance complex proteins 2, 3, and 5.** *Cancer Med* **11**, 4989-5000 (2022)), in which authors systematically reviewed and investigated the interaction of MCMs and wild-type and/or mutant p53s.

• *Related to this: why as a DeltaDBD (I assume a non-DNA binding version of p53) bind much more MCM5. This control (?) is not described in the manuscript (Fig 2C, lane 11), unless I overlooked it.*

Our response: Correct. Both non-DNA binding version and N-terminal deletion versions of mutp53 bound to MCM5 (Fig. 1d, lane 11 and 8), whereas C-terminal deletions lose MCM5-binding activities (Fig. 1d, lane 10), suggesting that mutp53 binds to MCM5 through its C-terminal domain. We included this information in text lines 139-142.

• *The authors present various Western blots in which RPA32 shows altered chromatin binding in p53 mutant cell lines and that this changes following overexpression of MCM5 or knockdown of MCM5 (in cell lines in which mutant p53 is knocked down). While the blots illustrate the findings well, the authors might want to quantify the effect and include multiple biological replicates to ensure this effect is consistent and to determine the average effect size.*

Our response: Thanks for this suggestion. In this revised manuscript, we have quantified all of the western blots, and quantification results are shown under each blot and original data are also provided in the Source Data. To ensure the consistency and accuracy of our results, all our major conclusions were based on the multiple independent western blots that have been shown in the paper. Here are examples:

1. For the impact of overexpression of mutp53 on RPA32 chromatin binding: Extended Data Fig.1h,i,k,l,o and Fig. 2a,b (total 7 independent western blots); 2. For the impact of downregulation of mutp53 on RPA32 chromatin binding: Fig. 2c and Extended Data Fig. 2a (2 blots),h (total 4 Western blots); 3. For the impact of MCM5 on RPA32 chromatin binding: Extended Data Fig.1k,l,o; Fig. 2b,c and Extended Data Fig. 2h (total 6 independent blots); 4. For the impact of overexpression of mutp53 on NC-NF-kB: Fig. 3e and Extended Data Fig. 3p,q (total 3 independent blots); 5. For the impact of downregulation of mutp53 on NC-NF-kB: Fig. 3f and Extended Data Fig. 3r,s (total 3 independent blots); 6. For the impact of MCM5 on NC-NF-kB: Fig. 3e,f and Extended Data Fig. 3q,s (total 4 independent blots); etc.

• *Related to this, in e.g. Fig 2b, the authors observe an extra RPA32 band, but only in the chromatin-bound fraction. What is this second band: a phosphorylation product?*

Our response: Yes, this second band represents phosphorylated RPA32 (we have added this information in the Fig. 2b). RPA32 is hyperphosphorylated upon DNA damage or replication stress by checkpoint kinases including ataxia telangiectasia mutated (ATM), ATM and Rad3-related (ATR), and DNA-dependent protein kinase (DNA-PK), and Phosphorylation of RPA32 occurs at serines 4, 8, and 33 (Wu, X. et al. (2005) *Oncogene* 24, 4728-35; Binz, S.K. et al. *DNA Repair (Amst)* 3, 1015-24; Nuss, J.E. et al. (2005) *Biochemistry* 44, 8428-3.). However, under the low concentration of HU (100-200 μ M) (HU has usually been used in the mM range to induce replication stress in traditional replication stress studies), this phosphorylation was only obvious in human normal epithelial hTERT HAK cl41 cells (Fig. 2b), but not in other head and neck cancer cells (e.g., UM-SCC-1, MDA1986 and PCI15B cells, Fig. 2a,c, Extended Data Fig.1h,i,k,l,o, and Extended Data Fig. 2a,h), suggesting different cells have varied sensitivities to HU.

Therefore, for evaluating replication stress under our experimental condition, we measured both phosphorylated and/or total chromatin-bound RPA32 in our studies.

- *Similarly, the authors show multiple IF experiments in which RPA32 is localized in granules (vs. homogenous nuclear localization). This effect should also be quantified for multiple cells and in biological replicates.*

Our response: We are deeply appreciative of this suggestion. However, as we stated that we used low concentrations of HU (μM not mM) to induce chronic replication stress. Unlike DNA-damaging agents and regular (high) doses of HU used in the traditional DNA repair/damage and replication stress studies, RPA32 foci formed under our condition are usually very subtle and hard to be well-defined, especially in cancer lines used in the studies (e.g., UM-SCC-1, MDA1886 and PCI15B cells). In our studies involved in image-related quantifications, we have tried our best to use a machine learning-based approach (e.g., Fig. 3 and Fig. 5) to achieve unbiased conclusions. In this case, given the subtlety of foci formation, we were unable to develop an algorithm to guide the computer to count the foci. We were also not confident that whether our bare eyes-assisted counting can ensure the accuracy and objectivity of the results or not. Therefore, the purpose of our IF experiments was just to provide a qualitative, rather than a quantitative, evidence to support our western blot data. Because of these reasons, we believe that the western blotting of extraction buffer-insoluble chromatin fractions (see the Methods) was the best and most objective way that could ensure the most accurate results without biases under the experimental conditions we used. Since western blots were well-monitored by their internal controls, we could easily draw our conclusions with high confidence. In this sense, our IF results actually were quantified in different way as the corresponding western results shown in the paper. For examples: Extended Data Fig. 1j was quantified by Fig. 2b; Extended Data Fig. 2b was quantified by Extended Data Fig. 2a and Extended Data Fig. 2i was quantified in Fig. 2c. Finally, we have also used DNA fiber assay to measure and quantify the replication stress in our study (Extended Data Fig. 2c,d).

- *The authors use HU to induce replication stress and determine cytoplasmic ssDNA and dsDNA. While they nicely show and quantify this using IF and DNA-recognizing antibodies, the authors do not quantify micronuclei, which is another important instigator of cGAS-STING signaling. As micronuclei are a common outcome of replication stress and their rupture will trigger cGAS, the authors should also quantify these in the various genotypes and following HU treatment.*

Our response: We greatly appreciate this suggestion. However, given the same reason discussed above, we were not sure how reliable to count micronuclei under our experimental conditions. In fact, in addition to computer-based quantification of IF using DNA antibodies (Fig. 3a-d, g-j and Extended Data Fig. 3c,d,g,h,k,l,m,n), we also directly measured cytosolic DNA (Extended Data Fig. 3e,i, j and the Methods). In addition, in this revised manuscript we also measured the intracellular cGAMP that is a direct indicator of cGAS activation (Extended Data Fig. 3o).

- *The authors quantify CIN using metaphase spreads and show clear effects. Adding time-lapse data of cells going through mitosis would be a complementary technique to strengthen this point and help determine whether the abnormalities in the metaphase spreads (partly) result from mitotic abnormalities that are secondary to the replication stress.*

Our response: We deeply appreciate your suggestion. However, we feel that whether the abnormalities in the metaphase spreads were partly resulted from mitotic abnormalities that are secondary to replication stress or not was not the primary focus of our current study. In addition, given the heterogeneity of tumor cells and the experimental conditions as we discussed above, we were not sure that how representative of a limited numbers of time-lapse data will be, although as pointed out by this reviewer that this technique could be complementary to and strengthen our points. Therefore, rather

than investigating the cause of mitotic abnormalities under the low level of replication stress, we have prioritized our studies on the biological consequences resulting from mutp53-induced CIN in our current study. Further investigation of mitotic abnormalities that are secondary to the replication stress associated with different p53 forms and their interactions with MCMs is something worth pursuing as a separate study.

- *Authors discuss the role of cGAS-STING pathway in the induction of NC-NF- κ B signaling and all the downstream effects. However, all the experiments relating to this pathway are done in the context of shSTING or shRelB. It has been previously shown that STING can be activated independently of cGAS. Therefore, authors might want to show the role of cGAS in their phenotype as well.*

Our response: This is an excellent point and in order to address this, we have established new cell lines and our results showed that cGAS, like Sting and RelB, is required not only for NC-NF- κ B activation but also for related increases in cell migration/invasion (Fig. 4a, g,h), suggesting that it is a cGAS-dependent process. This result is consistent with and supported by the latest publication from Dr. Bakhoun and colleagues (Li, J. *et al.* **Non-cell-autonomous cancer progression from chromosomal instability.** *Nature* **620**, 1080-1088 (2023)).

- *Where applicable, authors should more clearly show the individual replicate values for each experiment (i.e. Fig 4f, 4h, m, q, 5d,...)*

Our response: Yes, we have followed the journal instruction and all the quantification figures were shown as either dot plots or violin plots ($n > 7$). In violin plots, n was shown in the figure legend. Please also see the original numbers in Source Data for each figure panel.

- *In figure 4b, there is a clear increase in p53 expression upon knockdown of RelB. Why is this? The authors should comment on that.*

Our response: This is an interesting observation, and not only does RelB knockdown increase p53 levels [Fig. 4d (previously Fig. 4b) and 4e], but also cGAS and Sting knockdown tends to increase p53 levels (Fig. 4a-c). The reason for this remains to be determined. However, RelB has been previously shown to negatively regulate p53 stability through RelB direct targets PSMA5 [proteasome (prosome, macropain) subunit, a type, 5] and ANAPC1 [APC1, Anaphase Promoting Complex Subunit 1; a component of cycle-regulated E3 ubiquitin ligase Anaphase Promoting Complex (APC/C)], in which depletion of RelB increased p53 stability [Iannetti, A. *et al.* Regulation of p53 and Rb links the alternative NF- κ B pathway to EZH2 expression and cell senescence. *PLoS Genet* **10**, e1004642 (2014)]. Therefore, our observation of the negative regulation of p53 expression by RelB, cGAS or Sting (they are upstream of RelB) could be attributed to the same mechanisms as described above. More importantly, despite that mutp53 levels were increased after RelB, cGAS or Sting knockdown, cells with these increased mutp53 lost GOF activities (Fig. 4g-p), which further supports the importance of cGAS-Sting-RelB signaling in mutp53-associated GOF, and that GOF activities are actually not correlated with mutp53 expression levels as has been previously reported [(Redman-Rivera, L. N. *et al.* Acquisition of aneuploidy drives mutant p53-associated gain-of-function phenotypes. *Nat Commun* **12**, 5184 (2021)].

- *In figure 4c, a p53 blot is missing.*

Our response: We re-ran the western, and provided a new figure as presented in Fig. 4e (previously Fig. 4c). also see the above explanation.

- *In Figure 4, authors show invasion assays using R273H mut UM-SCC1 cells in combination with shRelB or shSTING. Authors also show similar results G245D mut UM-SSC-1 using shRelB. Authors should also show the effect of shSTING in invasion capacity of G245D mut UM-SSC-1 cells.*

Our response: Yes, we also established cell lines accordingly and the results are shown in Fig. 4c,k,l.

• *In Fig4k, there appears to be some mislabeling of the genotypes in the schematic representation. For instance, in "(5) R270H" should be replaced for "(5) R273H" and in "(6) R273H-shSting1" should be replaced for "(6) R273H-shRelB2"*

Our response: Thank you for pointing out our mislabeling errors. We have corrected them accordingly (Fig. 4q, previously Fig. 4k).

• *In Fig5f and 5g, authors should perform and include CD3 and CD8 quantification from at least 8 (out of the 11) primary 4T1 tumors. This will make their data more robust.*

Our response: We deeply appreciate the recommendation. Yes, it was indeed a challenge to objectively quantify IF staining. Traditionally, people just pick-up certain "random" areas in each section to count. However, given the heterogeneity of tumor tissue, we often found the staining was not homogeneous, which varied from areas to areas that are dependent on the section selected (e.g., tumor central areas vs peripheral areas, etc.). Therefore, any these kinds of quantification results are relatively "subjective" since it totally depends on the sections and/or areas selected. As described, in our studies we have relied our quantification on AI-based machine-learning (Extended Data Fig. 5b). To best avoid the "subjectivity" of our results, we have used entire tumor section (**not** just used certain areas in the section) to quantify for each sample. Take Fig. R1 (see below) as an example, we can see that this tumor section was quite heterogeneous. So, to achieve our quantification, we had to train the computer to first distinguish tumor areas, necrotic areas, normal mammary gland, etc., and then to phenotype and quantify the cells from entire tumor areas in this section (not just from certain areas) (see the Methods). In Fig. 5f,g, each data point actually represented the results from entire tumor areas from one tumor section that usually contained $0.3-1.7 \times 10^6$ nuclei or cells (see the Source Data to Fig. 5; Fig5 legend and Methods). We had a total of 4 different data points representing 4 individual tumor sections from 4 different tumors in each group. In short, we have utilized costly (in terms of time and money) machine-learning, and computational approaches, which limited further replicates. However, given the robustness of our approach as just described, we believe that compared to conventional IHC quantification used by many researchers, our results have already provided us with abundant, reliable data needed to come out our conclusions. Thank you very much!

Fig. R1: A example of H&E section of R270H mutp53 4T1 tumor whose adjacent section was used for multiplex IF staining and analyses.

• *In the transcriptome studies (Fig 6a, b), we miss RelB KO cells in a wildtype background (i.e. without p53 GOF mutations).*

Our response: Yes, to address this, we established a new cell line to knock down RelB in p53-null UM-SCC-1 cells (Extended Data Fig. 5d), and then we carried out an additional RNA-seq using these RelB-knockdown cells without any p53 GOF mutation. These data (UM-SCC-1 pBabe-shNT vs pBabe-shRelB1) have been incorporated into Fig. 6a, column 7 (also see the source data and Gene Expression Omnibus (GSE164433)). This new result again supports our observation that NC-NF- κ B inhibits IFN signaling. In addition, as we expected, the gene expression range of our latest sequencing results from pBabe-shNT vs pBabe-shRelB1 were significantly differed from that of the previous experiments due to different sequencing batch effects. In this way, the only way to plot the data (heatmap) was to calculate a z-score per sample (each gene receives a value considering the mean expression of all genes in that sample). However, our latest (pBabe-shNT and pBabe-shRelB1) transcriptome data contains only 13,380 genes, while our previous experiment had 23,473 genes (the reasons we just received 13,380 gene transcriptome data this time remain to be determined). This difference causes us a big variation in the mean expression value per sample, consequently deviating the z-score distribution between the 2 sequencing batches. For these reasons, we just included the latest pBabe-shNT vs pBabe-shRelB1 comparison results in Fig. 6a and chose NOT to incorporate the original expression data into heatmaps in Fig. 6b and Extended Data Fig. 6a. In spite of this, our new results will not affect our conclusion and they again demonstrated that NC-NF- κ B inhibits IFN signaling.

• *Using TCGA data, authors showed that tumors with low scores on IFN signaling but high inflammatory signatures have a lower NK cell activated signature and lowered follicular helper signature. Is this trend also observed in their vivo experiments?*

Our response: We examined activated NK cells in 4T1 stable tumors using multiplexed ion beam imaging (MIBI) and results showed that the signal of activated NK cell (CD49⁺CD3⁻Granzyme B⁺) was too

low for us to make a conclusion (data not shown). In spite of this, our results has shown that the numbers of Granzyme B⁺ (a member of the granule serine protease family specifically from NK and cytotoxic T cells) cells were indeed lower in mutp53-expressing tumors than that in the controls (Fig 5h,i). This has provided additional supporting evidence that mutp53 expression suppresses NK and/or T cell activities. In addition, the numbers of dendritic cells were also shown to be different from our MIBI studies (Fig. 5j,m). These new results (Fig. 5h-m) have been described in text lines 297-305. Finally, since we did not have a validated antibody panel for mouse Tfh maker (CD4, CXCR5, ICOS, and PD-1), Tfh cells in 4T1 tumors remain to be determined.

- *Could authors generalize their findings of 4 distinct groups of patients with IFN high/low or inflammation high/low OSCC to other cancer types with mutp53 using TCGA?*

Our response: To answer the reviewer's query regarding the relevance of our findings for other *TP53* mutant tumors, we conducted further research looking for similar molecular phenotypes (illustrated in Figure 6d of our manuscript) among other *TP53* mutant squamous tumors from TCGA. For this additional investigation, we selected cases of larynx and lung squamous cell carcinoma (LUSC), but excluded other squamous tumors from other anatomical sites. Oropharynx and cervical squamous cell carcinomas were excluded due to their primary association with HPV, while bladder and esophagus tumors were excluded due to their low number of *TP53* mutant cases, as well as significant molecular differences from HNSCC (see Figure 1 from Campbell et al. 2018 – Genomic, Pathway Network, and Immunologic Features Distinguishing Squamous Carcinomas. Cell Reports 23(1)194-212). The new Extended Data Fig. 8a-j present these new analyses indicating the presence of similar molecular patterns in *TP53* mutant larynx and lung squamous carcinomas. It is important to note that a trend toward high EMT signature and lower density of activated natural killer (NK) cells, CD8⁺ cells, and follicular helper T-cells was observed in tumors with the "IFN-low & Inf-high" molecular phenotype in both larynx and lung tumors. These findings suggest the existence of a similar molecular mechanism among squamous tumors that have *TP53* mutations. However, our study design does not allow us to draw definitive conclusions about other tumor types. Further studies are needed to confirm if these molecular processes are relevant for any *TP53* mutant carcinomas. See also text lines: 360-364.

- *The authors show that NC-NFKB inhibits IFN signaling. Understanding how this works mechanistically would add significantly to the impact of the study. If this is beyond the scope of this study, the authors should at least discuss how they think this works.*

Our response: Thank you very much! As we have discussed in the discussion, in which we have cited two papers (references 62 and 63) [(Hou, Y. et al. Non-canonical NF-kappaB Antagonizes STING Sensor-Mediated DNA Sensing in Radiotherapy. *Immunity* **49**, 490-503 e494 (2018); Jin, J. et al. The noncanonical NF-kappaB pathway controls the production of type I interferons in antiviral innate immunity. *Immunity* **40**, 342-354 (2014). See text line 407-408], in which authors showed that by suppressing recruitment of the transcription factor RelA onto the *Ifnb* promoter, activation of the non-canonical NF-κB pathway inhibited type I IFN expression. Therefore, it is possible that through the similar mechanism, NC-NF-κB inhibited IFN signaling as we have described.

We are very grateful for this reviewer's comments that have helped us to elevate the quality and impact of this manuscript.

Reviewer #3 (Remarks to the Author):

In the paper, the authors try to find the mechanism by which gain-of-function of mutant p53 proteins promotes tumor development and progression. After finding p53 interactors, the authors focus on MCM5

and show how the mutant p53-MCM5 interaction drives chromosomal instability which leads to STING-dependent response and activation of non-canonical NFkB signaling. These in turn promote tumor cell metastasis and an immunosuppressive tumor microenvironment. The authors investigate the chain of reactions leading to mutant p53 GOF phenotype in an orderly fashion, yet some information is missing and should be addressed by the authors as described below.

Immunoprecipitation and mass spectrometry analysis:

1. The authors use for the IP two different p53 antibodies: DO-1 and Pab240. According to the manufacturer's website, Pab240 does not bind WT p53 in its native form, only the mutant protein (<https://www.scbt.com/p/p53-antibody-pab-240>, "p53 Antibody (Pab 240) is recommended for detection of only mutant p53 under non-denaturing conditions but is equally reactive with mutant and wildtype p53 under denaturing conditions"). Is this difference in reactivity was taken into consideration in the experimental design? The authors should state these differences in the text and methods section.

Our response: Thanks for this insightful question. Yes, Pab240 does not bind WT p53 in its native form. The purpose of our study was to investigate the mutp53-interactome. As described in the manuscript, we stably introduced G245D mutp53 into **p53-null** UM-SCC-1 cells [UM-SCC-1 cell has splice site mutant rendering the cells null for p53 protein expression and resulting in loss of p53 function (Wilkie, M. D. et al. *TP53 mutations in head and neck cancer cells determine the Warburg phenotypic switch creating metabolic vulnerabilities and therapeutic opportunities for stratified therapies. Cancer Lett 478, 107-121 (2020)*]. Our previous studies showed that G245D mutp53 in UM-SCC-1 cells gains oncogenic functions to promote tumor cell migration/invasion (also see Fig. 4k-n and data not shown), especially in the presence of overexpression of PI3K mutations (data not shown, manuscript in preparation). Therefore, to investigate the possible mechanism involved, here we used two cell lines for IP assay: UM-SCC-1-mutp53G245D and UM-SCC-1-mutp53G245D & PIK3CA H1047R. Since UM-SCC-1 cells did not have any endogenous p53, the only p53 in stable cell lines was exogenously expressed G245D mutp53. In other words, when we used both DO-1 (against wt & mutp53) and Pab240 (against only mutp53) antibodies in our pull-down assay, all the p53 pulled down by IP were presumably the G245D mutant form. Therefore, we believe the difference of the antibody Pab240 in reactivity with WT/Mutant p53 was not an issue in our study.

2. Supplementary table 1 – please provide titles to the different experiments, it is not clear which sample is heavily labeled or not, and which cells were used or compared in each experiment.

Our response: Thanks for the suggestion. In the revised Supplementary Table 1, we have added new titles at the top of each column to show the cell lines and the reagents used to label the corresponding cell lines.

3. According to the methods section, the authors used for some of the IP experiments cells with PIK3CA H1047R mutation. It is only mentioned in the methods section and not explained why this mutation was included and how it affects the IP results.

Our response: Our previous study indicated that while G245D mutp53 gained oncogenic functions to promote UM-SCC-1 cell growth and cell invasion, expression of PIK3CA H1047R in the presence of G245D further enhanced cell growth and invasion activities of UM-SCC-1 cells (data not shown, manuscript in preparation). This information was included in Text lines 711-714. We initially wondered whether there are differences of mutp53 interactome either in the presence or the absence of expression constitutively active PI3K. However, our data did not give a clear indication, suggesting mutp53 and PI3K may function independently in UM-SCC-1 cells.

4. From the IP experiment design, it is not possible to know which interactors are preferentially binding to the mutant or WT p53 protein. Overexpressing each form separately and performing the IP can solve

it. In the IP experiment, both WT and mutant forms are probably immunoprecipitated (as it is not indicated that the endogenous WT p53 was knocked out). Alternatively, if one of the antibodies binds only to the mutant form maybe the authors can use it for showing the differential binding of interactors to the mutant compared to the WT p53.

Our response: As we discussed above, parental UM-SCC-1 cells used in IP-MS do not have endogenous p53 protein. Validation western blots of p53-null expression of this cell were also shown in: Extended Data Fig. 1h,i, lanes 1 & 2; Extended Data Fig. 1k, lanes 1-4, 9-12; Extended Data Fig. 1l, lanes 1-3, 9-12; Extended Data Fig. 1o, lanes 1-4; Fig. 3e, lanes 1-4; Extended Data Fig. 3a, lanes 1-2 and Extended Data Fig. 4d,e; lane 1). Therefore, the only form of p53 pulled-down in UM-SCC-1 stable cells was exogenously expressed mutp53s.

In addition, we have overexpressed each form of p53, including WT and mutp53s into cells to compare their interaction with MCM5 as shown in Fig. 1d.

5. What is the ratio of WT and mutant p53 protein expression in the cells used in the study? Making a table that includes the mutation/WT p53 status of each cell that was used in the study, as well as the relative protein abundance of the mutant/WT p53 protein and MCM5 protein, can help the readers understand the effect of their abundance on the observed phenotypes and make an easy to follow guide for the different cells.

Our response: As we mentioned previously, all cell lines used in our study **DO NOT** have WT p53, and they are all p53 mutated cell lines. The p53 status of each cell line used has been described in the text, corresponding western Figs, and the Methods section.

6. The authors should provide validation of protein overexpression in the cells, by western blot or any other means.

Our response: This is an excellent suggestion, and in all the western blots from Figs.1-4 and Extended Data Figs. 1-3, p53 expression and other related proteins have been shown and validated. For examples: hTERT HAK cl41-p53KO-c1 (p53 knockout) stable cells: Fig. 2a,b.

UM-SCC-1 (p53-null) stable cells: Extended Data Fig. 1f,h,i,k,l,o; Fig. 3e; ; Extended Data Fig. 3a,p,q; Fig. 4a-e and Extended Data Fig. 5d.

MDA1586 cells (endogenous R273L mutp53) stable cells: Fig. 1b,g; Fig. 2c; ; Extended Data Fig. 2a,f; ; Fig. 3f and Extended Data Fig. 3r.

PCI-15B cells (endogenous R273C mutp53) stable cells: Fig. 1c; Extended Data Fig. 2a,g,h and Extended Data Fig. 3s.

4T1 (p53-null) stable cells: Fig. 4f

7. The methods section is lacking a lot of information including how IP samples were prepared for mass spectrometry analysis, there is no information at all about the mass spectrometry analysis (which instrument? Which setting? And so on), what were the setting for the MaxQuant search? Which MaxQuant version was used? Did the mutation sequence was included in the database? How the data was processed and normalized?

Our response: We appreciate these comments. The details of Mass Spectrometry and software were updated in the methods of the revised manuscript.

8. Data deposition: The authors provided the user name but not the password for reviewing their data deposition in PRIDE. Please make sure to include in addition to the search and raw files also the fasta file that was used for the search, the MaxQuant installation file of the version that was used for analyzing the data, and a file that describes the raw file name with the sample name. Please provide in the revised

version of the manuscript the password for the data deposition so it could be accessed and reviewed.

Our response: Thanks for the comment.

The account and password to review PRIDE deposit data are:

Username: reviewer_pxd039660@ebi.ac.uk;

Password: gjwBwo1b

The detailed information of MaxQuant version and the database for search were updated in the revised manuscript. For the raw files, we have 8 samples, and each sample has 3 fractions (three raw data files). We added this information into the Sample name in Supplementary Table 1 in the revised manuscript.

IP results:

1. The authors identified 33 different proteins that interact with mutp53, but except for listing them, no further information is provided. Do the authors see any pathway/function enrichment for the interactors? Do other interactors were derived from protein complexes (except MCM proteins)? Which ones are already known and which are new interactions?

Our response: Thanks for the comments. We have added more previous publications to support our findings (text lines 122-124). We also run Metascape GO analysis that showed several pathways were enriched, including MCM5-involved GO term of regulation of DNA replication initiation (Fig. 1a and Text lines 125-128).

2. The authors should describe in the text why they decided to focus only on MCM5 out of all the interactors that were identified? What was the rationale behind their choice?

Our response: This is a very important point. First, as we shown in Fig. 1a and Source Data to Fig. 1a, GO terms of regulation of replication initiation was one of the top pathways enriched involving MCM5, MCM7 and WRNIP1 from our mass spectrometry results. Therefore, to further study the mechanism involved, we then focused our study on MCM5 and MCM7 (text lines 125-134). Second, after evaluating the binding activity with R273H mutp53, we found that although R273H/L and several other mutp53s interacted with different components of the MCM2-7 complex (e.g., MCM5, MCM2, MCM3, and MCM7), they bound most strongly to MCM5 (Fig. 1f,g and Extended Data Fig. 1c-e) (text lines: 143-145). Therefore, Given the strong interaction between MCM5 and R273H/L mutp53 (Fig. 1f, g and Extended Data Fig. 1f), we then focused our current study as an “proof-of-principle” study on investigation of the functional role of the interaction between MCM5 and R273H/L mutp53 in regulation of DNA replication, and then we further explored its consequential biological implications (text lines: 167-171).

3. Did the authors validate all unknown interactions as well (using WB or any other approach)?

Our response: No, so far, we still did not have the chances to validate all the “unknown interactions”. However, the references we provided in the revised manuscript (text lines 122-124), including previous publications of the interaction between MCM5/MCM7 and mutp53 (text line 123), have provided us high confidence in our results.

Validation of mutp53-MCM5 interaction:

1. For all WB images, the authors should provide also uncropped blots.

Our response: Every single uncropped blot for all western blots shown in this paper have been provided with the revised manuscript.

2. Why the validation of the interactions with MCM5 and other MCM (as in figure 1) were not done in the cells in which the IP experiments were performed (UM-SCC-1)?

Our response: Thanks for the comment. Previously, we prioritized the validation using cells with endogenous mutp53s. In this revised manuscript, IP validation results from UM-SCC-1 stable cells were

also provided (Extended Data Fig. 1f,g).

3. Figure 1 and Extended Figure 1:

a. Please indicate in the figure legend that p53 was KD in the cells as control (it is only indicated in the figure as shp53).

Our response: Thanks for the suggestion. We have included this information in the legends of Fig. 1 and Extended Data Fig. 1.

b. Why in some of the cells p53 was KD while other cells treated with IFN γ ?

Our response: Since we had mutp53 KD cells in MDA1586 and PCI15B cell lines (Fig. 1b,g and Extended Data Fig.1b), we used KD cells to better evaluate the IP specificity besides using IgG. In addition, IFN γ was previously shown to promote the interaction of MCM5 and Stat1 (DaFonseca, C. J., Shu, F. & Zhang, J. J. *Identification of two residues in MCM5 critical for the assembly of MCM complexes and Stat1-mediated transcription activation in response to IFN-gamma. Proc Natl Acad Sci U S A* **98**, 3034-3039 (2001) (this information has been included in text lines 130 and 134), which initially promoted us to investigate whether IFN γ has an impact on MCM5-mutp53 interaction.

4. Page 6: “although R273H/L and several other mutp53s also interacted with other components of the MCM2-7 complex (e.g., MCM2, MCM3, and MCM7), they bound most strongly to MCM5” – did the data normalized to each MCM expression level, can it be that the enrichment in pull down result from different MCM protein expression and not binding affinity?

Our response: This conclusion was based on co-IP results from Fig. 1f, in which relative binding (IP amount) was calculated based on the normalized same level of different MCM protein expression (See the Source Data to Fig. 1f for details). Therefore, the enrichment in pull down was not resulted from different MCM protein expression level.

Other comments:

1. Page 4: “This finding strongly suggests that many previously reported mutp53 GOF phenotypes are actually consequences of CIN in mutp53 cells” – this statement should be accompanied by analyzing previous data to show it, or the text should be adjusted to suggest that this is the author's assumption.

Our response: Thanks for the suggestion. We have adjusted our text to: “This finding strongly suggests that the consequences of CIN in mutp53 cells may contribute to many previously reported mutp53 GOF phenotypes”. See text lines 84-86.

2. Page 5: “Here, we examined the mutp53 interactome and identified MCM5 as a direct covalently bound protein target through which mutp53 exerts its GOF activities to predispose cancer cells with mutp53 to replication stress and CIN” – the authors do not demonstrate in the study that MCM5 and mutp53 are bound covalently or that MCM5 and mutp53 bind directly. The IP experiments in the manuscript do not prove any of these claims, please adjust the text or perform the required experiments to prove it.

Our response: We appreciate and agree with reviewer’s comments. We have adjusted the text to: “MCMs as the protein target through which”, Text lines 103-106.

3. Why TCGA analysis was done only for HPV-negative OSCC patients? do the observations in the study are relevant only to this cancer type and not for any mutant p53 cancer types?

Our response: We used HPV-negative and TP53-mutant OSCC for our initial analysis. Additional analyses in HPV-negative and TP53-mutant lung and larynx squamous cell carcinomas were also included in the revised manuscript (Extended Data Fig. 8a-j and text lines: 360-364).

Head and neck squamous cell carcinomas (HNSCC) can be divided into two distinct entities based on the presence or absence of human papillomavirus (HPV) infection. HPV-negative HNSCC is typically associated with tobacco and ethanol use, whereas HPV-related HNSCC affects younger patients, affects the oropharynx, and is highly responsive to radiation-based therapy. HPV-mediated tumors are mostly wild-type *TP53* (e.g., 97% HPV-positive HNSCCs are wild-type *TP53*), and the cell's defensive apoptotic mechanisms are inhibited by the virus through the promotion of p53 protein degradation. In this way, p53-related signaling pathways are remarkably distinct between these two tumor entities. Due to significant biological and clinical differences, HPV-positive and HPV-negative HNSCC tumors should not be considered as a homogeneous group in molecular studies.

To answer the reviewer's query regarding the relevance of our findings for other *TP53* mutant tumors, we conducted further research looking for similar molecular phenotypes (illustrated in Figure 6d of our manuscript) among other *TP53* mutant squamous tumors from TCGA. For this additional investigation, we selected cases of larynx and lung squamous cell carcinoma (LUSC), but excluded other squamous tumors from other anatomical sites. Oropharynx and cervical squamous cell carcinomas were excluded due to their primary association with HPV, while bladder and esophagus tumors were excluded due to their low number of *TP53* mutant cases, as well as significant molecular differences from HNSCC (see Figure 1 from Campbell et al. 2018 – Genomic, Pathway Network, and Immunologic Features Distinguishing Squamous Carcinomas. Cell Reports 23(1)194-212).

The Extended Data 8 presents these new analyses indicating the presence of similar molecular patterns in *TP53* mutant larynx and lung squamous carcinomas. It is important to note that a trend toward higher EMT score and lower density of activated NK cells, CD8+ cells, and follicular helper T-cells was observed in tumors with the "IFN-low & Inf-high" molecular phenotype in both larynx and lung tumors. These findings suggest the existence of a similar molecular mechanism among squamous tumors that have *TP53* mutations. However, our study design does not allow us to draw definitive conclusions about other tumor types. Further studies are needed to confirm if these molecular processes are relevant for any *TP53* mutant carcinoma.

Please also see our response to Reviewer #4 for the similar question why we initially selected HPV-negative OSCC patients.

We are extremely appreciative of this reviewer's many insightful questions and points that we have addressed, thereby improving our manuscript.

Reviewer #4 (Remarks to the Author):

In the article 'Mutant p53 gains oncogenic functions through a chromosomal instability-induced cytosolic DNA response' the authors look at how gain-of-function mutants are linked to CIN in various cancer cells. They identified a role for mutant p53 in binding MCM5 and promoting a cytosolic DNA accumulation that promotes the STING pathway and suppresses the IFN release. The manuscript is well-written and the figures are clear. Mutant p53 has been associated to Chromosomal instability and in this manuscript the authors are addressing a novel pathway through which mutant p53 acts on this. This work is therefore a nice novel addition to previous work by others.

Although each individual figure is clear and draws the right conclusions, the work using the patient data does not really support the first part of the paper in which mutant p53 is regulating replication stress through MCM5 binding. These data merely address whether an interferon response is correlated the NF- κ B activity and survival, but do not take into account mutant p53 status or MCM5 activity. Statistics and

the use of error bars (SD or SEM) in the paper are a bit random and leave some questions.

Our response: We greatly appreciate these comments, and would like to highlight that while none of the data from patients have been linked to p53 status or MCM5 expression (the reasons for not including p53 status or MCM5 are discussed in detail in the following section), we used the INF low/inf high signature derived from our cell model to further explore the biological insights into the genome guardian role of p53 and its mutations in development of CIN that leads to metastasis and immunosuppression (See the following responses in detail).

Major comments:

1. *The authors select for patients with mutant p53 status. All mutations or only point mutants? What mutants are selected? A 248Q mutations is relatively frequent in OSCC and should not activate this pathway as that one does not bind MCM5. If selecting only 248Q patients is this actually the case? Are there enough patients to do this in the OSCC groups?*

Our response: This is an excellent question also raised by the reviewer #2. Here are our responses:

(1) R248Q mutations is also a GOF mutation that interacts with MCM7 and predisposes cells to replication stress:

As we discussed above (the response to the reviewer #2), in our original mass spectrum data, in addition to MCM5 we also pulled out MCM7 (supplementary table 1). Despite that our current study mainly focused on the MCM5's role given its strong interaction with R273H mutp53, we provided herein additional evidence showing that although R248Q mutp53 did not bind to MCM5 well (Fig. 1d, lane 7 and Extended Data Fig. 1f, lane 6), it strongly bound to MCM7 (Extended Data Fig. 1g, lane 4). In addition, R248Q mutp53 exhibited a greater potential to induce replication stress (Extended Data Fig. 1h, lanes 6 vs 4) and *in vitro* transwell invasion (data not shown) than did R273H mutp53 in UM-SCC-1 cells despite that its expression level was lower than that of R273H mutp53 (Extended Data Fig. 1h, lanes 5 & 6 vs 3 & 4). Therefore, through the same mechanisms (i.e., mutp53-MCMs-CIN-cGAS-Sting-NC-NF- κ B signaling) by interacting with and targeting different members of MCM2-7 subunits such as MCM7, R248Q mutp53 is expected to predispose cells to CIN and achieve the GOF activities as we described in this study. Our current study using R273H-MCM5 was an example of a "proof-of-principle" study that demonstrated a novel mechanism involved in mutp53 gain-of-functions. We have included all these new information in texts lines 145-149, 159-164, 167-171, 438-452. Therefore, based on all our results, we have generalized our hypothesis to MCMs and not only restrict our hypothesis to MCM5 (see Abstract and Fig. 6k).

(2) The clinical implications of R248Q and other TP53 mutations in TCGA

In our initial analysis, we selected all the HPV(-) and TP53 mutant OSCC in TCGA for our analysis (Fig. 6). In this revised manuscript, HPV(-) and TP53 mutant larynx and lung squamous carcinomas were also included (Extended Data Fig. 8 and text lines: 360-364)[(please also see our response to the last question of reviewer#3 regarding the biology difference between HPV(+) and HPV(-) tumors)]. In all these cases, p53 mutations selected included any kinds of mutations including missense point mutation, truncation, frame-shift mutation, deletion, etc.

The reviewer's question regarding the p53 mutant status in TCGA is an excellent question related to how we think the fundamental role of p53 mutations. Yes, it is well-documented that patients with wtp53 tumors have better survival than patient with mutant p53s, including the OSCC as we primarily focused on here in our current study (since we are at the department of head and neck surgery, and head and neck cancers are our primary focus). However, it is indeed a great challenge to further correlate mutant p53 status with patient data, especially regarding so called "GOF mutations" vs "LOF mutations" in TCGA. There has been a big debate whether there is a true GOF of mutant p53s or not. Despite that abundant *in vitro* and *in vivo* evidence, including our current study, support the concept of GOF mutp53 activities, others have suggested that there are no actual differences among different p53 mutations vs

mutp53 (see the discussion section of Redman-Rivera, L. N. *et al.* Acquisition of aneuploidy drives mutant p53-associated gain-of-function phenotypes. *Nat Commun* **12**, 5184 (2021)). However, one consensus so far in the p53 field is that GOF activities of mutp53s are context-dependent which explains why there are so many conflicting results in the literature (see the introduction section).

The importance of our current study, together with the recent study from Dr. Pietenpol and colleagues, is that we demonstrate that GOF activities of mutp53s are driven by mutp53-induced CIN rather than mutp53 expression level, and that there is no causal relationship between mutp53 expression and mutp53 GOF activities (even different subclones in the same cell line with the similar level of the same mutp53 expression may behave differently (Redman-Rivera, L. N. *et al.* Acquisition of aneuploidy drives mutant p53-associated gain-of-function phenotypes. *Nat Commun* **12**, 5184 (2021)). Our results strongly suggest that CIN itself (like cGAS-Sting-NC-NF- κ B signaling as we described here) and the consequence of CIN (many altered genes and pathways resulting from CIN) may contribute to observed GOF activities of mutp53s. In this sense, we believe the biggest lesson from our studies is that there are no fundamental differences between “GOF mutations” and “LOF mutations” since all inactivating p53 mutations lead to CIN. The only difference between GOF and LOF mutations may be that GOF mutations (or hotspot mutations?) have a greater propensity to target MCM2-7 complex and predispose cells to CIN than do LOF mutations, which enable GOF mutations (e.g., hotspot mutations) to gain selective advantage during tumor development and progression.

Interestingly, despite the many studies, including ours, using *in vitro* cell lines and *in vivo* animal models have demonstrated the GOF properties of mutp53s, there may be no easy ways in cancer patients to tell which p53 mutations are worse than others in terms of clinical outcome. Take the hotspot R248 mutation as an example, as the reviewer has pointed out. In our total 221 HPV (-) OSCC tumors with mutant p53s (all kinds of mutations), there are 5.4% (12/221) R248 mutants, their distributions are: 9% in IFN-low/inf-high group; 6% in IFN-high/inf-low group; 3% in IFN-low/inf-low group and 3% IFN-high/inf-high group. Despite that it, appeared with a little higher frequency in IFN-low/inf-high group than that in others (9% vs 6% or 3%), we are still not sure what is the biological significance or difference of this hotspot R248 mutation in terms of its impact on clinical outcomes when compared to other p53 mutations in this cohort (no differences were observed, data not shown). Similarly, we also did not see neither the differences of other hotspot mutations across the different groups nor the difference between the hotspot mutations and other non-hotspot mutations in terms of their clinical outcomes in this cohort (data not shown). Therefore, the relationship of mutp53s and clinical data is much more complex than we thought.

However, the reason of these complexities in the patients may become clearer given our current studies. Since both the results from Dr. Pietenpol and colleagues [(Redman-Rivera, L. N. *et al.* Acquisition of aneuploidy drives mutant p53-associated gain-of-function phenotypes. *Nat Commun* **12**, 5184 (2021))] and our current study clearly demonstrate that CIN is the key for mutp53 GOF. Therefore, it is not hard to imagine that both CIN and altered genes/signaling pathways resulting from CIN all contribute to adverse clinical outcomes. Because the processes and the outcomes of developing CIN induced by both GOF mutp53s and LOF mutp53s are **RANDOM**, it is easy for us to understand why **NO** direct correlations of the expression of certain mutations (e.g., R248 or other hotspot mutations) with the clinical outcomes in TCGA patients were observed. For all these reasons, we did not link the mutant p53 status in our analyses or we did not see any experimentally testable correlation of certain specific p53 mutations with adverse clinical outcomes. Instead, we took all mutant p53 patients as a group and used the INF low/inf high signature derived from our cell model to explore the biological insights into the in-depth mechanisms involved.

Finally, MCMs are usually very abundant in all kinds of the cells and many MCM components were shown overexpressed in different cancers. However, the expression levels of MCM5 or other MCM members are not relevant to our current study. Therefore, we did not link their expression to our TCGA analyses neither in our study.

2. Does a loss of Sting in 248Q cells still result in a RelB depletion in the nucleus (fig 4A)? To prove that the Sting pathway is downstream the mutant p53/MCM5 interaction

Our response: The answer should be YES. A loss of cGAS-Sting signaling in any kinds of cells regardless of p53 status should impact RelB (also see Bakhoum, S. F. *et al.* Chromosomal instability drives metastasis through a cytosolic DNA response. *Nature* **553**, 467-472 (2018). As we discussed above, R248Q binds to MCM7, predisposing cells to replication stress that also activates RelB through cGAS-Sting.

3. RPA32 is used as a marker of replication stress. The authors look in the chromatin fraction and have indicated both bands as markers for RPA32 binding to chromatin. I presume the higher band is p-RPA32, which many in the field use as marker for replication stress. Clearly this phospho-band is following the pattern as described in the results section for figure 2A and 2B. The bottom RPA-32 band does not and stays relatively equal in fig 2B. In Figure 2C no top band is seen and the authors make the conclusions about the bottom band. Can the authors clarify they are looking at both bands and reference that binding to chromatin regardless of phosphorytion of RPA-32 is a marker of replication stress.

Our response: Yes, the reviewer is right. We used both *phosphorylated RPA-32* and total chromatin-bound RPA as the marker for replication stress. As we also responded to the Reviewer #2, the higher band represents phosphorylated RPA32 (we have added this information in the Fig. 2b). RPA32 is hyperphosphorylated upon DNA damage or replication stress by checkpoint kinases including ataxia telangiectasia mutated (ATM), ATM and Rad3-related (ATR), and DNA-dependent protein kinase (DNA-PK), and Phosphorylation of RPA32 occurs at serines 4, 8, and 33 (Wu, X. *et al.* (2005) *Oncogene* 24, 4728-35; Binz, S.K. *et al.* *DNA Repair (Amst)* 3, 1015-24; Nuss, J.E. *et al.* (2005) *Biochemistry* 44, 8428-3.). However, under the low concentration of HU (100-200 μ M) (HU has usually been used in the mM range to induce replication stress in traditional replication stress studies), this phosphorylation was only obvious in human normal epithelial hTERT HAK cl41 cells (Fig. 2b), but not in other head and neck cancer cells (e.g., UM-SCC-1, MDA1986 and PCI15B cells, Fig. 2a,c, Extended Data Fig.1h,i,k,l,o, and Extended Data Fig. 2a,h), suggesting different cells have varied sensitivities to HU. Therefore, for evaluating replication stress under our experimental condition, we measured phosphorylated and/or total chromatin-bound RPA32 in our studies.

4. The authors use a lot of different cell lines and it is not always clear why certain cell lines are used for different purposes. Some clarification is needed in the text

Our response: The cell lines used with their mutant p53 status have been listed in the Methods (text lines 646-654). As we described in text lines 167-171, our current study was primarily focused on the R273H-MCM5 relationship. Therefore, in addition to p53-null cell lines (i.e., hTERT HAK cl41-p53KO and UM-SCC-1) used to establish mutp53 stable cells, other cell lines we chose are with R273 mutations (i.e., MDA1586, PCI15B).

5. In figure 4m the authors show a bar graph with very small error bars. The supplemental data show that some mice did not develop any metastases. It therefore seems very unlikely such small error bars could be achieved if taken into account the numbers of mice not developing any metastases. More than half of the mice in the R270H shRelB 1 didn't develop mets. The graph shows 10% with an error bar of less than 1%. Surely half of these datapoints would have been 0 (no mets). To then get to 10% the

remaining 5 mice must have had an area of 20% or more with mets to get to a 10% mean (suggested in legend) which would result in very large error bars. Statistics is unclear

Our response: Yes, the reviewer is right. As shown in Extended Data Fig. 4b and the Source Data to Fig. 4s (previously Fig. 4m), there were indeed several mice in each group that did not develop lung metastasis during the **18-weeks'** experimental time (see Figure 4 legend), which resulted in a much larger error bar as the reviewer pointed out. The 0 metastasis was related to the nature of low/medium metastatic potential of UM-SCC-1 cells. In addition, we wish we could have extended our experimental time longer than **18-weeks** (see Fig 4 legend) to make sure every mouse developed lung metastasis. But our experiment was forced to be terminated a little bit earlier than what we expected due to the beginning of Covid-19 pandemic break that led to the restriction of our animal facility at that time. However, despite of all these, if we just use "yes or no" as the criteria, our results from both macroscopic and microscopic metastases were completely consistent with and supported our conclusion about the role of the mutp53-Sting-RelB signaling in metastasis (Extended Data Fig. 4b).

Fig. 4s (previous Fig. 4m) is our further attempt to present our results with a meaningful "statistical value", in which we scored the % metastatic areas from 3 independent thick H & E sections (100 to 300 μm apart, the traditional consecutive sections usually are only 2-3 μm apart). Then we took one section from each mouse to form a set and calculate its mean % metastatic areas (8-10 mice/set). The results shown in Fig. 4s (previous Fig. 4m) were from 3 sets (shown as 3 points), each point represented the mean value from 8-10 mice (see the Source Data to Fig. 4s). We have clarified this in the Y-axis title of Fig. 4s, and please also see Figure 4 legend for details.

Although this approach of identifying microscopic metastasis is not perfect, given above reasons and especially given another layer of the complexity that it was not possible for us to cut through the entire tumor to identify more microscopic metastases in each tumor, we believe that this approach, together with directly counting macroscopic metastases (Extended Data Fig. 4b), were the best way we could come to a conclusion that objectively reflect the metastatic potentials of the cell lines we investigated under our experimental condition.

Finally, due to the similar reasons, the same approach was also applied in Fig. 4w and Extended Data Fig. 4o.

6. Error bars in most figures are very small despite large variations seen in raw data and between samples. Mainly concerning in the in vivo data. Are the authors sure these depict SD and not SEM? Very confusing that different error bars are used in different figures and even parts of figures. Why different statistics test in 6g and h compared to I and J in the same figure?

Our response: SEM was only used in one Figure (Fig. 5a) to better visualize the tumor growth. SD was used in all the rest Figures in the manuscript. The original data for all figures shown in this manuscript have been provided in the Source Data.

As we just discussed above (our response to question #5), the small error bars of our in vivo data regarding the microscopic metastasis (e.g. Fig. 4s,w and Extended Data Fig. 4o) were mainly resulted from the particular quantification approach we chose under our experimental conditions. We have clarified our approach in the Y-axis title and Figure legends accordingly. We admit that our approach of identifying lung microscopic metastasis is not perfect given the nature of low/medium metastatic potential of UM-SCC-1 cells (e.g., Fig. 4s, 18 weeks after tail-vein injection, still not every mouse developed lung metastasis) or given the short time duration of 4T1 orthotopic tumor model (Fig. 4w, we had to sacrifice the mice only 21-day after mammary gland injection due to the large tumor volumes of the primary tumors, which limited 4T1 tumor lung metastases)(see the Source data). However, we believe that our approach of counting the microscopic metastases, together with directly counting macroscopic metastases (e.g. Extended Data Fig. 4b and Fig. 4v), were the best way we could come to a

conclusion that objectively reflect the metastatic potentials of the cell lines we investigated under our experimental condition.

Regarding figures 6g-h: The Kruskal-Wallis test was used to assess the overall difference among the 4 groups, followed by the Steel-Dwass test to assess differences between group pairs. Figures 6g-h and 6i-j show identical statistics.

7. MCM5 itself is causing chromosomal instability and ssDNA and dsDNA accumulation in the cytoplasm. However, in the presence of mutant p53, it inhibits the mutant p53 mediated accumulation. It is not explained how this works.

Our response: Thank you for this insightful question. As we described in text lines 183-189, overexpression of MCM5 alone is likely to disrupt the stoichiometry of MCM2-7 complex, impair its function (i.e., dominant-negative effect) and result in replication stress. In fact, In the UM-SCC-1 (p53-null) cells, we could not get a cell line with high levels of MCM5 expression. After MCM5 viral infection, most of cells exhibited senescence phenotypes that are consistent with typical cGAS-STING-induced senescence, and eventually died (data not shown). The surviving cells had a marginal overexpression of MCM5 (e.g. Fig. 3e...cytosol fraction lanes 3 & 4 vs 1 &2). However, in the presence mutp53, excess MCM5 binds to mutp53 making more MCM5 available for functional MCM complexes and thereby inhibiting mutp53-mediated replication stress.

We thank this reviewer for the many helpful questions and suggestions which have given us the chance to make this paper stronger.

REVIEWERS' COMMENTS

Reviewer #1 (Remarks to the Author):

The authors have done a great job addressing my comments in this revised manuscript. I think this work will be of interest to the general cancer community and I support publication.

Reviewer #2 (Remarks to the Author):

The authors have addressed the key issues in their revised manuscript. While they did not follow up all suggestions, they justified their choices well (some of my suggestions were somewhat out of the main scope) and therefore I feel that the manuscript is now suitable for publications. Well done!

Reviewer #3 (Remarks to the Author):

The authors have completely addressed my concerns.

A few comments for data and information completion:

1. The authors should include in the text a few details regarding the experiments (as provided in their answers in the reviewer comments), so they will be easily available for the readers of the paper :

* Please add in the methods section regarding UM-SCC-1 cells that they are p53-null.

* Please add in the methods section why you selected to use UM-SCC-1-mutp53G245D & PIK3CA H1047R cells. You can note that it is part of a manuscript in preparation.

2. Please include in the PRIDE deposition the FASTA file with the proteome that was used for MS analysis. As well as the MaxQuant Application (.exe), of the version that was used for analyzing the data (you can zip and upload it). Different MaxQuant versions can give varied results, and this will allow others to reproduce your data.

Reviewer #4 (Remarks to the Author):

The authors have added an impressive amount of new data and corrected any ambiguities in the previous version. The additional work in combination with the new publication by Li et al makes this a much stronger manuscript. Inconsistencies with the 248Q can be explained by the new data implicating other MCM proteins. No further corrections needed

Dear Reviewers:

We greatly appreciate your comments and your support to publish this important work. Thank you very much!

Here are our additional responses:

REVIEWERS' COMMENTS

Reviewer #1 (Remarks to the Author):

The authors have done a great job addressing my comments in this revised manuscript. I think this work will be of interest to the general cancer community and I support publication.

Our response: Thank you very much!

Reviewer #2 (Remarks to the Author):

The authors have addressed the key issues in their revised manuscript. While they did not follow up all suggestions, they justified their choices well (some of my suggestions were somewhat out of the main scope) and therefore I feel that the manuscript is now suitable for publications. Well done!

Our response: Thank you very much!

Reviewer #3 (Remarks to the Author):

The authors have completely addressed my concerns.

A few comments for data and information completion:

1. The authors should include in the text a few details regarding the experiments (as provided in their answers in the reviewer comments), so they will be easily available for the readers of the paper :

* Please add in the methods section regarding UM-SCC-1 cells that they are p53-null.

Our response: Thank you very much. This information has been added in the method section text lines 463-465.

* Please add in the methods section why you selected to use UM-SCC-1-mutp53G245D & PIK3CA H1047R cells. You can note that it is part of a manuscript in preparation.

Our response: Thank you very much. This information has been added in the method section text lines 525-529.

2. Please include in the PRIDE deposition the FASTA file with the proteome that was used for MS analysis. As well as the MaxQuant Application (.exe), of the version that was used for analyzing the data (you can zip and upload it). Different MaxQuant versions can give varied results, and this will allow others to reproduce your data.

Our response: Thanks for the suggestions. We have updated our PRIDE deposition with the FASTA file and MaxQuant Application as suggested. Please check the updated deposition with the new PRIDE project accession number PXD047094 with username "reviewer_pxd047094@ebi.ac.uk" and password "9aHK6OeT". We also updated our manuscript with the new accession number.

Now, we have also released our data to the public:

<https://proteomecentral.proteomexchange.org/cgi/GetDataset?ID=PXD047094>

Thank you very much!

Reviewer #4 (Remarks to the Author):

The authors have added an impressive amount of new data and corrected any ambiguities in the previous version. The additional work in combination with the new publication by Li et al makes this a much stronger manuscript. Inconsistencies with the 248Q can be explained by the new data implicating other MCM proteins. No further corrections needed

Our response: Thank you very much!